# Controlling Multiple Errors Simultaneously with a PAC-Bayes Bound

**Reuben Adams**
Department of Computer Science
University College London
reuben.adams.20@ucl.ac.uk

**John Shawe-Taylor**
Department of Computer Science
University College London
j.shawe-taylor@ucl.ac.uk

**Benjamin Guedj**
Department of Computer Science, University College London and Inria
b.guedj@ucl.ac.uk

## Abstract

Current PAC-Bayes generalisation bounds are restricted to scalar metrics of performance, such as the loss or error rate. However, one ideally wants more information-rich certificates that control the entire distribution of possible outcomes, such as the distribution of the test loss in regression, or the probabilities of different mis-classifications. We provide the first PAC-Bayes bound capable of providing such rich information by bounding the Kullback-Leibler divergence between the empirical and true probabilities of a set of $M$ error types, which can either be discretized loss values for regression, or the elements of the confusion matrix (or a partition thereof) for classification. We transform our bound into a differentiable training objective. Our bound is especially useful in cases where the severity of different mis-classifications may change over time; existing PAC-Bayes bounds can only bound a particular pre-decided weighting of the error types. In contrast our bound implicitly controls all uncountably many weightings simultaneously.

## 1 Introduction

Generalisation bounds are a core component of the theoretical understanding of machine learning algorithms. For over two decades now, PAC-Bayesian theory has been at the core of studies on generalisation abilities of machine learning algorithms. PAC-Bayes originated in the seminal work of McAllester [1998, 1999] and was further developed by Catoni [2003, 2004, 2007], among other authors—we refer to the surveys Guedj [2019] and Alquier [2021] for an introduction to the field. The outstanding empirical success of deep neural networks in the past decade calls for better theoretical understanding of deep learning, and PAC-Bayes has emerged as one of the few frameworks that can be used to derive meaningful (and non-vacuous) generalisation bounds for neural networks: the pioneering work of Dziugaite and Roy [2017] has been followed by a number of contributions, including Neyshabur et al. [2018], Zhou et al. [2019], Letarte et al. [2019], Pérez-Ortiz et al. [2021], Perez-Ortiz et al. [2021] and Biggs and Guedj [2022a,b], to name but a few.

Much of the PAC-Bayes literature focuses on the case of binary classification, or of multiclass classification where one only distinguishes whether each classification is correct or incorrect. This is in stark contrast to the complexity of contemporary real-world learning problems, such as medical diagnosis where the severity of Type I and Type II errors may be crucial and context-dependent. This work aims to bridge this gap by deriving a generalisation bound that provides information-rich measures of performance at test time by controlling the probabilities of errors of any finite number of user-specified types. More precisely, we bound the KL-divergence between the empirical and true

distributions over the different error types. From this single bound one can then derive bounds on arbitrary linear combinations of these error probabilities, which will all hold simultaneously with the same probability as the original bound. In addition, these bounds are guaranteed to be non-vacuous (this follows since the KL-divergence blows up on the boundary of the simplex).

As a concrete example, if the severity of Type I and Type II errors of a medical test are context-dependent, one would want to be able to bound arbitrary linear combinations of these error probabilities. Existing bounds could only bound finitely many pre-specified weightings by employing a union bound, which would also degrade the bound. In contrast, by constraining the KL-divergence between the true and empirical error probabilities, our bound constrains all uncountably many weightings of the error probabilities simultaneously.

The usual setting of PAC-Bayes bounds is that of binary classification, namely an input set $\mathcal{X}$, output set $\mathcal{Y} = \{-1, 1\}$, hypothesis space $\mathcal{H} \subseteq \mathcal{Y}^{\mathcal{X}}$ and a sample $S \in (\mathcal{X} \times \mathcal{Y})^m$ drawn i.i.d. from a data-generating distribution $D$. A number of PAC-Bayes bounds in this setting (e.g. Maurer [2004]) have been unified by a single general bound found in Bégin et al. [2016]. Briefly, Bégin et al. [2016] prove a bound on the discrepancy $d(R_S(Q), R_D(Q))$ between the error probability $R_D(Q)$ of a stochastic classifier $Q$ (a distribution over $\mathcal{H}$ which classifies by first drawing $h \sim Q$ and then classifying according to $h$) and its empirical counterpart $R_S(Q)$ (the fraction of the sample $Q$ misclassifies). The bound holds with high probability for all $Q$ simultaneously. The bound in Bégin et al. [2016] is binary in the sense that $\mathcal{Y}$ contains two elements, but a more subtle way to look at this is that only two cases are distinguished—correct classification and incorrect classification. While it can be applied to multiclassification provided one maintains the second binary characteristic by only distinguishing correct and incorrect classifications. It is this heavy restriction that our result lifts, by considering the new framework of *error types*.

**A new framework of errors types**   We consider a user-specified partition of the space $\mathcal{Y} \times \mathcal{Y}$ of prediction-truth label-pairs into a finite partition of error types $E_1, \ldots, E_M$. Our bound then simultaneously constrains the probability with which errors of each type occur. In multiclass classification for example, one can choose the error types to be the set of all different possible mis-classifications, in which case our bound will control the entire confusion matrix, bounding how far the true confusion matrix (i.e. expected over the data-generating distribution) can diverge from the empirical one (i.e. on the training set). From this one can then derive bounds on the probabilities with which each mis-classification may be made, and arbitrary linear combinations of these error probabilities, and all of these will hold simultaneously with the same probability as the original bound. Our bound therefore paints a far richer picture of the performance of the final learned model than can be provided by any existing PAC-Bayes bound.

Formally, we let $\bigcup_{j=1}^M E_j$ be a user-specified disjoint partition of $\mathcal{Y}^2$ into a finite number of $M$ *error types*, where we say that a hypothesis $h \in \mathcal{H}$ makes an error of type $j$ on datapoint $(x, y)$ if $(h(x), y) \in E_j$ (by convention, every pair $(\hat{y}, y) \in \mathcal{Y}^2$ is interpreted as a predicted value $\hat{y}$ followed by a true value $y$, in that order). It should be stressed that not all of the $E_j$ need correspond to mislabellings—indeed, some of the $E_j$ may distinguish different correct labellings.

**Relation to previous results.**   Our framework of a finite number of user-specified "error types" includes multiclass classification as a particular case, and it is in this field that one finds the work most closely related to ours. Little is known of multiclass classification from a theoretical perspective and, to the best of our knowledge, only a handful of relevant strategies or generalisation bounds can be compared to the present paper.

The closest is the work of Morvant et al. [2012], which establishes a PAC-Bayes bound on the spectral norm of the difference between the true and empirical confusion matrices. Our bound differs from theirs in two respects. First, they consider the confusion matrix, whereas ours applies to the more general setting of a finite number of error types, which can be the set of all mis-classifications, or some partition thereof. Second, they deal with the spectral norm, whereas we employ the KL-divergence. Since the KL-divergence follows a simple formula, this means we can much more easily infer bounds on the individual error probabilities, which would be very challenging for the spectral norm. The follow-up work Koço and Capponi [2013] shows how a proxy of the spectral norm bound can be used as a training objective that may deal with imbalanced classes. In the present work, we show how our bound can be used as a differentiable training objective directly (without the need of a proxy) and

that it can more sensitively deal with imbalanced classes, or errors of different severity, by assigning each error type a user-specified loss value.

Laviolette et al. [2017] extend the celebrated $\mathcal{C}$-bound in PAC-Bayes to ensembles, obtaining a bound on the risk of the majority vote classifier in the case of multiclass classification. In this context, our bound is able to distinguish different mis-classifications and control them, whereas they bound the scalar risk which lumps all mis-classifications together. The $\mathcal{C}$-bound has alternately been generalised by Lacasse et al. [2006] (see also Germain et al. [2015]) to simultaneously control three metrics, namely the so-called *expected disagreement, expected joint success and expected joint error* of the posterior. While they restricted themselves to the ternary case, some of their proof techniques share similarities with ours. In cases where one has exactly three error types, for example the $\{-1, 0, 1\}$-valued *excess loss*, the work of Wu and Seldin [2022] is applicable; they construct so-called 'split-kl' inequalities (both classical and PAC-Bayesian) which deftly handle this specific scenario.

Pires et al. [2013] present a comprehensive analysis of convex surrogate losses in cost-sensitive multiclass classification, providing conditions for consistency, bounding the excess loss of a predictor, and extending the analysis to the "Simplex Coding" scheme. We are considering the generalisation gap rather than the excess loss. Lei et al. [2019] study data-dependent bounds for multiclass classification. Their analysis is restricted to SVMs however, whereas ours applies to arbitrary hypothesis spaces.

**Outline.** We fix notation in Section 2. Theorem 1 in Section 3 is our main result—a PAC-Bayes bound on the KL-divergence between the true and empirical error distributions. For multiclass classification with a fully refined partition this becomes a bound on the KL-divergence between the true and empirical confusion matrices. Proposition 1 then bounds the individual error probabilities. Our second main result, Theorem 2 in Section 4, allows us to use bounds on *linear combinations* of error probabilities as training objectives. We prove Theorem 1 in Section 5 via Proposition 4, which bounds the distribution of errors via a general convex function $d$ and may be of independent interest. Section 6 outlines positive empirical results[1] from using our bound as a training objective for neural networks and Section 7 gives perspectives for follow-up work..

## 2 Notation

For any set $A$, let $\mathcal{M}(A)$ be the set of probability measures on $A$. Let $\mathcal{X}$ and $\mathcal{Y}$ be arbitrary input (*e.g.*, feature) and output (*e.g.*, label) sets respectively, and $D \in \mathcal{M}(\mathcal{X} \times \mathcal{Y})$ be a data-generating distribution. For any sample $S \sim D^m$ drawn i.i.d. from $D$, let $\hat{D}(S) \in \mathcal{M}(\mathcal{X} \times \mathcal{Y})$ denote the empirical distribution $\hat{D}(S) := \frac{1}{m} \sum_{(x,y) \in S} \delta_{(x,y)}$. We consider the setting where the user has specified a partition $\{E_1, \ldots, E_M\}$ of $\mathcal{Y}^2$ into $M$ *error types*.

We are interested in *simple* hypotheses $h : \mathcal{X} \to \mathcal{Y}$ and *soft* hypotheses $H : \mathcal{X} \to \mathcal{M}(\mathcal{Y})$. For example, a neural network outputting scores (logits) in $\mathbb{R}^{\mathcal{Y}}$ is converted to a simple or soft hypothesis, respectively, by passing the scores through the argmax or softmax function, respectively. For any $A \subseteq \mathcal{Y}$, $H(x)(A)$ can be interpreted as the probability according to $H$ that the label of $x$ is in $A$. We will see in Section 4 that soft hypotheses permit more flexible training procedures and a more fine-grained analysis. Note that while soft hypotheses output distributions, they do so deterministically, always returning the same distribution for the same input $x$, and so are distinct from the stochastic classifiers introduced shortly.

For a simple hypothesis $h : \mathcal{X} \to \mathcal{Y}$ and $j \in [M]$, define the *j-risk* of $h$ to be $R_D^j(h) := \mathbb{P}_{(x,y) \sim D}((h(x), y) \in E_j)$, namely the probability that $h$ makes an error of type $E_j$ for a randomly sampled $(x, y) \sim D$. For a soft hypothesis $H : \mathcal{X} \to \mathcal{M}(\mathcal{Y})$ define the *j-risk* of $H$ to be $R_D^j(H) := \mathbb{P}_{(x,y) \sim D, \hat{y} \sim H(x)}((\hat{y}, y) \in E_j)$, namely the probability that one would make an error of type $E_j$ on a randomly sampled $(x, y) \sim D$ if one predicted by sampling $\hat{y}$ from the distribution $H(x)$. From now until Section 4 it will not matter whether we are dealing with simple or soft hypotheses. So, unless stated explicitly, we will refer to both simply as hypotheses, denote both by lowercase $h$, and refer to the hypothesis class $\mathcal{H}$, whether it is a subset of $\mathcal{Y}^{\mathcal{X}}$ or $\mathcal{M}(\mathcal{Y})^{\mathcal{X}}$.

Our goal is to control the *risk vector* $\boldsymbol{R}_D(h) := (R_D^1(h), \ldots, R_D^M(h))$, since controlling this vector controls all linear combinations of $j$-risks. Since this is unobervable, we will control it by

---

[1]Code available here: `https://github.com/reubenadams/PAC-Bayes-Control`

bounding how far it diverges from its empirical counterpart $\boldsymbol{R}_S(h) := \boldsymbol{R}_{\hat{D}(S)}(h)$, which we term the *empirical risk vector*. Note that $\mathbb{E}_{S \sim D_m} \boldsymbol{R}_S(h) = \boldsymbol{R}_D(h)$, and that, for a simple hypothesis $h \in \mathcal{Y}^{\mathcal{X}}$, $\boldsymbol{R}_S(h)$ is the vector of proportions of the sample on which $h$ makes an error of type $E_j$[2]. Since the $E_j$ partition $\mathcal{Y}^2$, $\boldsymbol{R}_D(h)$ and $\boldsymbol{R}_S(h)$ are elements of the $M$-dimensional simplex $\triangle_M := \{\boldsymbol{u} \in [0,1]^M : u_1 + \cdots + u_M = 1\}$. Thus we can choose our divergence measure to be $\mathrm{kl}(\boldsymbol{R}_S(h) \| \boldsymbol{R}_D(Q))$, where for $\boldsymbol{q}, \boldsymbol{p} \in \triangle_M$ we define $\mathrm{kl}(\boldsymbol{q} \| \boldsymbol{p}) := \sum_{j=1}^M q_j \ln \frac{q_j}{p_j}$.[3] When $M = 2$ we abbreviate $\mathrm{kl}((q, 1-q) \| (p, 1-p))$ to $\mathrm{kl}(q \| p)$, which is then the conventional definition of $\mathrm{kl}(\cdot \| \cdot)$ found in the PAC-Bayes literature [as in Seeger, 2002, for example]. We define the *risk* and *empirical risk* of $Q$ as $\boldsymbol{R}_D(Q) := \mathbb{E}_{h \sim Q} \boldsymbol{R}_D(h)$ and $\boldsymbol{R}_S(Q) := \mathbb{E}_{h \sim Q} \boldsymbol{R}_S(h)$, respectively, and seek a bound on $\mathrm{kl}(\boldsymbol{R}_S(Q) \| \boldsymbol{R}_D(Q))$. Note we still have $\mathbb{E}_S[\boldsymbol{R}_S(Q)] = \boldsymbol{R}_D(Q)$, this time using Fubini. Moreover, for a sample $S$ of size $m$, we have that $\boldsymbol{R}_S(Q) = \boldsymbol{K}/m$ where $\boldsymbol{K} \sim \mathrm{Mult}(m, M, \boldsymbol{R}_D(Q))$. Recall that for $m, M \in \mathbb{N}$ and $\boldsymbol{r} \in \triangle_M$, the multinomial distribution $\mathrm{Mult}(m, M, \boldsymbol{r})$ has probability mass function $\mathrm{Mult}(\boldsymbol{k}; m, M, \boldsymbol{r}) := \binom{m}{k_1 \ k_2 \ \cdots \ k_M} \prod_{j=1}^M r_j^{k_j}$, where $\binom{m}{k_1 \ k_2 \ \cdots \ k_M} := \frac{m!}{\prod_{j=1}^M k_j!}$ for $\boldsymbol{k} \in S_{m,M} := \{(k_1, \ldots, k_M) \in \mathbb{N}_0^M : k_1 + \cdots + k_M = m\}$, and zero otherwise. As a final piece of notation, we let $\triangle_M^{>0} := \triangle_M \cap (0,1)^M$ and $S_{m,M}^{>0} := S_{m,M} \cap \mathbb{N}^M$ denote the vector elements of $\triangle_M$ and $S_{m,M}$, respectively, that have no zero components.

# 3  Main result

We now state our main result, which bounds the KL-divergence between the true and empirical risk vectors $\boldsymbol{R}_D(Q)$ and $\boldsymbol{R}_S(Q)$, interpreted as probability distributions. As is conventional in the PAC-Bayes literature, we refer to sample independent and dependent distributions on $\mathcal{M}(\mathcal{H})$ (*i.e.* stochastic hypotheses) as *priors* (denoted $P$) and *posteriors* (denoted $Q$) respectively, even if they are not related by Bayes' theorem.

**Theorem 1.** *Let $\mathcal{X}$ and $\mathcal{Y}$ be arbitrary sets and $\bigcup_{j=1}^M E_j$ be a disjoint partition of $\mathcal{Y}^2$ into $M$ error types. Let $D \in \mathcal{M}(\mathcal{X} \times \mathcal{Y})$ be a data-generating distribution and $\mathcal{H}$ be a simple ($\mathcal{H} \subseteq \mathcal{Y}^{\mathcal{X}}$) or soft ($\mathcal{H} \subseteq \mathcal{M}(\mathcal{Y})^{\mathcal{X}}$) hypothesis class. For any prior $P \in \mathcal{M}(\mathcal{H})$, $\delta \in (0,1]$ and sample size $m \geq M$, with probability at least $1 - \delta$ over the random draw $S \sim D^m$, we have that simultaneously for all posteriors $Q \in \mathcal{M}(\mathcal{H})$, the divergence $\mathrm{kl}(\boldsymbol{R}_S(Q) \| \boldsymbol{R}_D(Q))$ is upper bounded by*

$$\frac{1}{m} \left[ \mathrm{KL}(Q \| P) + \ln \frac{\xi(m, M)}{\delta} \right], \quad \text{where} \tag{1}$$

$$\xi(m, M) := \sqrt{\pi} e^{1/(12m)} \left(\frac{m}{2}\right)^{\frac{M-1}{2}} \sum_{z=0}^{M-1} \binom{M}{z} \left(\frac{2}{m}\right)^{z/2} \Gamma\left(\frac{M-z}{2}\right)^{-1} \in \mathcal{O}\left((mM)^M\right).$$

The fact that the logarithmic term is of order $\mathcal{O}(M \ln(mM/\delta))$ means the bound is linear in $M$ up to logarithmic terms, while this may seem excessive, one should note that the quantity that our theorem bounds also depends on $M$. Further, the bound has been successfully used in by Biggs and Guedj [2023] to improve on state of the art PAC-Bayes bounds.

To see how our bound compares to existing PAC-Bayes bounds for binary classification, take $\mathcal{Y} = \{-1, 1\}$, $M = 2$, $E_1 = \{(-y, y) : y \in \mathcal{Y}\}$ and $E_2 = \{(y, y) : y \in \mathcal{Y}\}$. The argument of the logarithm then reduces to $\frac{1}{\delta} e^{1/(12m)} \left(2 + \sqrt{\frac{\pi m}{2}}\right) \leq 1.25\sqrt{m}$ when $m$ is large. The corresponding term in Maurer [2004] is $2\sqrt{m}$, which is only larger because he relaxes the term for aesthetics. Therefore our bound gracefully reduces to Maurer's in the case of binary classification.

Suppose after a use of Theorem 1 we have a bound of the form $\mathrm{kl}(\boldsymbol{R}_S(Q) \| \boldsymbol{R}_D(Q)) \leq B$. We can then derive bounds on the individual $j$-risks $R_D^j(Q)$ or, more generally, on linear combinations thereof. While one could obtain such bounds perhaps more directly with existing PAC-Bayes bounds, the significance of our bound is that *all* such derived bounds hold with high probability *simultaneously*. Existing PAC-Bayes bounds would require the use of a union bound in order to bound multiple combinations simultaneously, whereas ours bounds all uncountably many combinations simultaneously, as a package. As for the individual $j$-risks $R_D^j(Q)$, the following proposition

---

[2]$(\boldsymbol{R}_S(h))_j = R_{\hat{D}}^j(h) = \mathbb{P}_{(x,y) \sim \hat{D}}((h(x), y) \in E_j) = \frac{1}{m} \sum_{(x,y) \in S} \mathbb{1}[(h(x), y) \in E_j]$.

[3]We follow the usual convention that $0 \ln \frac{0}{x} = 0$ for $x \geq 0$ and $x \ln \frac{x}{0} = \infty$ for $x > 0$.

then yields the bounds $L_j \leq R_D^j(Q) \leq U_j$, where $L_j := \inf\{p \in [0,1] : \mathrm{kl}(R_S^j(Q)\|p) \leq B\}$ and $U_j := \sup\{p \in [0,1] : \mathrm{kl}(R_S^j(Q)\|p) \leq B\}$. Moreover, since in the worst case we have $\mathrm{kl}(\boldsymbol{R}_S(Q)\|\boldsymbol{R}_D(Q)) = B$, the proposition shows that the lower and upper bounds $L_j$ and $U_j$ are the tightest possible, since if $R_D^j(Q) \notin [L_j, U_j]$ then $\mathrm{kl}(R_S^j(Q)\|R_D^j(Q)) > B$ implying $\mathrm{kl}(\boldsymbol{R}_S(Q)\|\boldsymbol{R}_D(Q)) > B$. For a more precise version of this argument and a proof of Proposition 1, see Appendix C.4.

**Proposition 1.** *Let $\boldsymbol{q}, \boldsymbol{p} \in \triangle_M$. Then $\mathrm{kl}(q_j\|p_j) \leq \mathrm{kl}(\boldsymbol{q}\|\boldsymbol{p})$ for all $j \in [M]$, with equality when $p_i = \frac{1-p_j}{1-q_j} q_i$. for all $i \neq j$.*

More generally, suppose we can quantify how costly an error of each type is by means of a loss vector $\boldsymbol{\ell} \in [0,\infty)^M$, where $\ell_j$ is the loss we attribute to an error of type $E_j$. We may then be interested in bounding the *total risk* $R_D^T(Q) := \boldsymbol{\ell} \cdot \boldsymbol{R}_D(Q)$. Then, given a bound $\mathrm{kl}(\boldsymbol{R}_S(Q)\|\boldsymbol{R}_D(Q)) \leq B$ from Theorem 1, we can deduce

$$R_D^T(Q) \leq \sup\{\boldsymbol{\ell} \cdot \boldsymbol{r} : \boldsymbol{r} \in \triangle_M, \ \mathrm{kl}(\boldsymbol{R}_S(Q)\|\boldsymbol{r}) \leq B\} = \boldsymbol{\ell} \cdot \mathrm{kl}_{\boldsymbol{\ell}}^{-1}(\boldsymbol{R}_S(Q)|B), \qquad (2)$$

where we define $\mathrm{kl}_{\boldsymbol{\ell}}^{-1}(\boldsymbol{u}|c) \in \triangle_M$ as follows. To see that it is indeed well-defined (at least when $\boldsymbol{u} \in \triangle_M^{>0}$), see the discussion at the beginning of Appendix C.5.

**Definition 1.** *For $\boldsymbol{u} \in \triangle_M, c \in [0,\infty)$ and $\boldsymbol{\ell} \in [0,\infty)^M$, define $\mathrm{kl}_{\boldsymbol{\ell}}^{-1}(\boldsymbol{u}|c)$ to be an element $\boldsymbol{v} \in \triangle_M$ solving the constrained optimisation problem*

$$\textit{Maximise:} \quad f_{\boldsymbol{\ell}}(\boldsymbol{v}) := \boldsymbol{\ell} \cdot \boldsymbol{v}, \qquad (3)$$

$$\textit{Subject to:} \quad \mathrm{kl}(\boldsymbol{u}\|\boldsymbol{v}) \leq c. \qquad (4)$$

This motivates the following training procedure: search for a posterior $Q$ for which the bound $\boldsymbol{\ell} \cdot \mathrm{kl}_{\boldsymbol{\ell}}^{-1}(\boldsymbol{R}_S(Q)|B)$ on the total risk $R_D^T(Q)$ is minimised. While this requires a particular choice of loss vector $\boldsymbol{\ell}$, we emphasise that at the end of training, Theorem 1 bounds $\mathrm{kl}(\boldsymbol{R}_S(Q)\|\boldsymbol{R}_D(Q))$ and so can be used to bound *any* linear combination of the $j$-risks, not just the one given the loss vector $\boldsymbol{\ell}$ chosen for training. It is this flexibility which is the main advantage of our bound; changes in the severity of different error types over time do not require union bounds or retraining.

In the next section we provide a theorem for calculating $\mathrm{kl}_{\boldsymbol{\ell}}^{-1}(\boldsymbol{u}|c)$ and its derivatives so that the training procedure can be executed.

## 4 Construction of a Differentiable Training Objective

We now state and prove Theorem 2, which provides a speedy method for approximating $\mathrm{kl}_{\boldsymbol{\ell}}^{-1}(\boldsymbol{u}|c)$ and its derivatives to arbitrary precision, provided $c > 0$ and $\forall j\ u_j > 0$. The only approximation step required is that of approximating the unique root of a continuous and strictly increasing scalar function. Thus, provided the $u_j$ themselves are differentiable, Theorem 1 combined with Theorem 2 shows that the upper bound on the total risk can be used as a tractable and fully differentiable training objective. See Appendix A for more details, including a pseudocode algorithm and an implementation. Since the proof of Theorem 2 is rather long and technical, we defer it to Appendix C.5. The requirement that the $\ell_j$ are not all equal only rules out trivial cases where $R_D^T(Q)$ is independent of $\boldsymbol{R}_D(Q)$.

**Theorem 2.** *Fix $\boldsymbol{\ell} \in [0,\infty)^M$ such that not all $\ell_j$ are equal, and define $f_{\boldsymbol{\ell}} : \triangle_M \to [0,\infty)$ by $f_{\boldsymbol{\ell}}(\boldsymbol{v}) := \sum_{j=1}^M \ell_j v_j$. For all $\tilde{\boldsymbol{u}} = (\boldsymbol{u}, c) \in \triangle_M^{>0} \times (0,\infty)$, define $\boldsymbol{v}^*(\tilde{\boldsymbol{u}}) := \mathrm{kl}_{\boldsymbol{\ell}}^{-1}(\boldsymbol{u}|c) \in \triangle_M$ and let $\mu^*(\tilde{\boldsymbol{u}}) \in (-\infty, -\max_j \ell_j)$ be the unique solution to $c = \phi_{\boldsymbol{\ell}}(\mu)$, where $\phi_{\boldsymbol{\ell}} : (-\infty, -\max_j \ell_j) \to \mathbb{R}$ is given by $\phi_{\boldsymbol{\ell}}(\mu) := \ln(-\sum_{j=1}^M \frac{u_j}{\mu + \ell_j}) + \sum_{j=1}^M u_j \ln(-(\mu + \ell_j))$, which is continuous and strictly increasing. Then $\boldsymbol{v}^*(\tilde{\boldsymbol{u}}) = \mathrm{kl}_{\boldsymbol{\ell}}^{-1}(\boldsymbol{u}|c)$ is given by*

$$\boldsymbol{v}^*(\tilde{\boldsymbol{u}})_j = \frac{\lambda^*(\tilde{\boldsymbol{u}}) u_j}{\mu^*(\tilde{\boldsymbol{u}}) + \ell_j} \quad \textit{for } j \in [M], \quad \textit{where} \quad \lambda^*(\tilde{\boldsymbol{u}}) = \left(\sum_{j=1}^M \frac{u_j}{\mu^*(\tilde{\boldsymbol{u}}) + \ell_j}\right)^{-1}. \qquad (5)$$

*Further, defining $f_{\boldsymbol{\ell}}^* : \triangle_M^{>0} \times (0,\infty) \to [0,\infty)$ by $f_{\boldsymbol{\ell}}^*(\tilde{\boldsymbol{u}}) := f_{\boldsymbol{\ell}}(\boldsymbol{v}^*(\tilde{\boldsymbol{u}}))$, we have that*

$$\frac{\partial f_{\boldsymbol{\ell}}^*}{\partial u_j}(\tilde{\boldsymbol{u}}) = \lambda^*(\tilde{\boldsymbol{u}})\left(1 + \ln \frac{u_j}{\boldsymbol{v}^*(\tilde{\boldsymbol{u}})_j}\right) \qquad \textit{and} \qquad \frac{\partial f_{\boldsymbol{\ell}}^*}{\partial c}(\tilde{\boldsymbol{u}}) = -\lambda^*(\tilde{\boldsymbol{u}}). \qquad (6)$$

A final wrinkle in evaluating our bound is that while the empirical risk vector $\boldsymbol{R}_S(Q) = \mathbb{E}_{h \sim Q} \boldsymbol{R}_S(h)$ does not depend on the data-generating distribution $D$, the expectation over $Q$ may still be intractable. This would be the default case when $Q$ is a Gaussian over the weights of a multi-layer perceptron, for example. In such cases, we can estimate $\boldsymbol{R}_S(Q)$ via a Monte Carlo sample $\boldsymbol{R}_S(\hat{Q}) := \frac{1}{N} \sum_{n=1}^N \boldsymbol{R}_S(h_n)$ (where the $h_n$ are drawn i.i.d. from $Q$) and use the following two results. Proposition 2 shows that the $\mathrm{kl}(R_S^j(\hat{Q}) \| R_D^j(Q))$ can be simultaneously bounded, whence Proposition 3 can be used to obtain a bound on $\mathrm{kl}(\boldsymbol{R}_S(\hat{Q}) \| \boldsymbol{R}_D(Q))$.

**Proposition 2.** *Let $\boldsymbol{X} \sim \mathrm{Multinomial}(N, M, \boldsymbol{p})$. Then for any $\delta \in (0, 1)$, with probability at least $1 - \delta$ we have that for all $j \in [M]$ simultaneously $\mathrm{kl}\left(\frac{1}{N} X_j \| p_j\right) \leq \frac{\ln \frac{2M}{\delta}}{N}$.*

*Proof.* Each bound holds separately with probability at least $1 - \delta/M$ by Theorem 2.5 in Langford and Caruana [2001]. They then hold simultaneously by application of a union bound. $\square$

**Proposition 3.** *Suppose $\boldsymbol{q}, \boldsymbol{p}, \hat{\boldsymbol{q}} \in \triangle_M$ are such that $\mathrm{kl}(\boldsymbol{q} \| \boldsymbol{p}) \leq B_1$ and $\mathrm{kl}(\hat{q}_j \| q_j) \leq B_2$ for all $j \in [M]$. For each $j$, define $\underline{q}_j = \inf\{r \in [0, 1] : \mathrm{kl}(\hat{q}_j \| r) \leq B_2\}$. Then*

$$\mathrm{kl}(\hat{\boldsymbol{q}} \| \boldsymbol{p}) \leq M B_2 - \sum_{j=1}^M (1 - \hat{q}_j) \ln \frac{1 - \hat{q}_j}{1 - \underline{q}_j} + B_1 \max_j \frac{\hat{q}_j}{\underline{q}_j} \to B_1 \quad as \quad B_2 \to 0. \tag{7}$$

*Proof.* Deferred to C.1. $\square$

The fact that the bound on $\mathrm{kl}(\hat{\boldsymbol{q}} \| \boldsymbol{p}) \to B_1$ as $B_2 \to 0$ ensures that as we increase the size of our Monte Carlo sample for estimating $\boldsymbol{R}_S(Q)$ the bound on $\mathrm{kl}(\boldsymbol{R}_S(\hat{Q}) \| \boldsymbol{R}_D(Q))$ approaches that of $\mathrm{kl}(\boldsymbol{R}_S(Q) \| \boldsymbol{R}_D(Q))$, meaning in the limit we pay an arbitrarily small price in the bound for the approximation.

## 5 Proof of the main bound

We split the proof of Theorem 1 into three parts. First, we prove Proposition 4, a bound on $d(\boldsymbol{R}_S(Q), \boldsymbol{R}_D(Q))$ for an arbitrary convex function $d$, which may be of independent interest. Second, we prove Corollary 1 by specialising Proposition 4 to the case $d(\cdot, \cdot) = \mathrm{kl}(\cdot \| \cdot)$. Finally, we show that the bound in Theorem 1 is a loosened version of the bound in Corollary 1.

**Proposition 4.** *Let $d : \triangle_M^2 \to \mathbb{R}$ be jointly convex. In the setting of Theorem 1,*

$$d\big(\boldsymbol{R}_S(Q), \boldsymbol{R}_D(Q)\big) \leq \frac{1}{\beta} \left[ \mathrm{KL}(Q \| P) + \ln \frac{\mathcal{I}_d(m, \beta)}{\delta} \right], \quad where \tag{8}$$

$$\mathcal{I}_d(m, \beta) := \sup_{\boldsymbol{r} \in \triangle_M} \left[ \sum_{\boldsymbol{k} \in S_{m,M}} \mathrm{Mult}(\boldsymbol{k}; m, M, \boldsymbol{r}) \exp\left(\beta d\left(\frac{\boldsymbol{k}}{m}, \boldsymbol{r}\right)\right) \right].$$

This is a generalisation of the unifying PAC-Bayes bound given in Bégin et al. [2016] where we replace the scalar risk quantities $R_S(Q)$ and $R_D(Q)$ with their vector counterparts $\boldsymbol{R}_S(Q)$ and $\boldsymbol{R}_D(Q)$. To see this, note that we can recover it by setting $\mathcal{Y} = \{-1, 1\}$, $M = 2$, $E_1 = \{(-y, y) : y \in \mathcal{Y}\}$ and $E_2 = \{(y, y) : y \in \mathcal{Y}\}$. Then, for any convex function $d : [0, 1]^2 \to \mathbb{R}$, apply Theorem 4 with the convex function $d' : \triangle_M^2 \to \mathbb{R}$ defined by $d'((u_1, u_2), (v_1, v_2)) := d(u_1, v_1)$ so that Theorem 4 bounds $d'\big(\boldsymbol{R}_S(Q), \boldsymbol{R}_D(Q)\big) = d\big(R_S^1(Q), R_D^1(Q)\big)$ which equals $d(R_S(Q), R_D(Q))$ in the notation of Bégin et al. [2016]. Further, $\sum_{\boldsymbol{k} \in S_{m,2}} \mathrm{Mult}(\boldsymbol{k}; m, 2, \boldsymbol{r}) \exp\left(\beta d'\left(\frac{\boldsymbol{k}}{m}, \boldsymbol{r}\right)\right) = \sum_{k=0}^m \mathrm{Bin}(k; m, r_1) \exp\left(\beta d\left(\frac{k}{m}, r_1\right)\right)$, so that the supremum over $r_1 \in [0, 1]$ of the right hand side equals the supremum over $\boldsymbol{r} \in \triangle_2$ of the left hand side, which, when substituted into (8), yields the bound given in Bégin et al. [2016].

To prove Proposition 4 we require the following two lemmas. The first is the well-known change of measure in equality (Csiszár, 1975, Donsker and Varadhan, 1975). The second is a generalisation from Binomial to Multinomial distributions of a result found in Maurer [2004], the proof of which we defer to Appendix C.2.

**Lemma 1.** *For any set $\mathcal{H}$, any $P, Q \in \mathcal{M}(\mathcal{H})$ and any measurable function $\phi : \mathcal{H} \to \mathbb{R}$, $\underset{h \sim Q}{\mathbb{E}}\, \phi(h) \leq$ $\mathrm{KL}(Q\|P) + \ln \underset{h \sim P}{\mathbb{E}} \exp(\phi(h))$.*

**Lemma 2.** *Let $\boldsymbol{X}_1, \ldots, \boldsymbol{X}_m$ be i.i.d $\triangle_M$-valued random vectors with mean $\boldsymbol{\mu}$ and suppose that $f : \triangle_M^m \to \mathbb{R}$ is convex. If $\boldsymbol{X}_1', \ldots, \boldsymbol{X}_m'$ are i.i.d. $\mathrm{Mult}(1, M, \boldsymbol{\mu})$ random vectors, then $\mathbb{E}[f(\boldsymbol{X}_1, \ldots, \boldsymbol{X}_m)] \leq \mathbb{E}[f(\boldsymbol{X}_1', \ldots, \boldsymbol{X}_m')]$.*

The consequence of Lemma 2 is that the worst case (in terms of bounding $d(\boldsymbol{R}_S(Q), \boldsymbol{R}_D(Q))$) occurs when $\boldsymbol{R}_{\{(x,y)\}}(h)$ is a one-hot vector for all $(x, y) \in S$ and $h \in \mathcal{H}$, namely when $\mathcal{H} \subseteq \mathcal{M}(\mathcal{Y})^{\mathcal{X}}$ only contains hypotheses that, when labelling $S$, put all their mass on elements $\hat{y} \in \mathcal{Y}$ that incur the same error type[4]. In particular, this is the case for hypotheses that put all their mass on a single element of $\mathcal{Y}$, equivalent to the simpler case $\mathcal{H} \subseteq \mathcal{Y}^{\mathcal{X}}$ as discussed in Section 2. Thus, Lemma 2 shows that the bound given in Proposition 4 cannot be made tighter only by restricting to such hypotheses.

*Proof.* (of Proposition 4) The case $\mathcal{H} \subseteq \mathcal{Y}^{\mathcal{X}}$ follows directly from the more general case by taking $\mathcal{H}' := \{h' \in \mathcal{M}(\mathcal{Y})^{\mathcal{X}} : \exists h \in \mathcal{H} \text{ such that } \forall x \in \mathcal{X} \ h'(x) = \delta_{h(x)}\}$, where $\delta_{h(x)} \in \mathcal{M}(\mathcal{Y})$ denotes a point mass on $h(x)$. For the general case $\mathcal{H} \subseteq \mathcal{M}(\mathcal{Y})^{\mathcal{X}}$, using Jensen's inequality with the convex function $d(\cdot, \cdot)$ and Lemma 1 with $\phi(h) = \beta d(\boldsymbol{R}_S(h), \boldsymbol{R}_D(h))$, we see that for all $Q \in \mathcal{M}(\mathcal{H})$

$$
\begin{aligned}
\beta d\big(\boldsymbol{R}_S(Q), \boldsymbol{R}_D(Q)\big) &= \beta d \left( \underset{h \sim Q}{\mathbb{E}}\, \boldsymbol{R}_S(h), \ \underset{h \sim Q}{\mathbb{E}}\, \boldsymbol{R}_D(h) \right) \\
&\leq \underset{h \sim Q}{\mathbb{E}}\, \beta d\big(\boldsymbol{R}_S(h), \boldsymbol{R}_D(h)\big) \\
&\leq \mathrm{KL}(Q\|P) + \ln \left( \underset{h \sim P}{\mathbb{E}} \exp\Big( \beta d\big(\boldsymbol{R}_S(h), \boldsymbol{R}_D(h)\big) \Big) \right) \\
&= \mathrm{KL}(Q\|P) + \ln(Z_P(S)),
\end{aligned}
$$

where $Z_P(S) := \mathbb{E}_{h \sim P} \exp\big(\beta d(\boldsymbol{R}_S(h), \boldsymbol{R}_D(h))\big)$. Note that $Z_P(S)$ is a non-negative random variable, so that by Markov's inequality $\underset{S \sim D^m}{\mathrm{P}} \left( Z_P(S) \leq \frac{\mathbb{E}_{S' \sim D^m} Z_P(S')}{\delta} \right) \geq 1 - \delta$. Thus, since $\ln(\cdot)$ is strictly increasing, with probability at least $1 - \delta$ over $S \sim D^m$, we have that simultaneously for all $Q \in \mathcal{M}(\mathcal{H})$

$$
\beta d\big(\boldsymbol{R}_S(Q), \boldsymbol{R}_D(Q)\big) \leq \mathrm{KL}(Q\|P) + \ln \frac{\underset{S' \sim D^m}{\mathbb{E}} Z_P(S')}{\delta}. \tag{9}
$$

To bound $\mathbb{E}_{S' \sim D^m} Z_P(S')$, let $\boldsymbol{X}_i := \boldsymbol{R}_{\{(x_i, y_i)'\}}(h) \in \triangle_M$ for $i \in [m]$, where $(x_i, y_i)'$ is the $i$'th element of the dummy sample $S'$. Noting that each $\boldsymbol{X}_i$ has mean $\boldsymbol{R}_D(h)$, define the random vectors $\boldsymbol{X}_i' \sim \mathrm{Mult}(1, M, \boldsymbol{R}_D(h))$ and $\boldsymbol{Y} := \sum_{i=1}^m \boldsymbol{X}_i' \sim \mathrm{Mult}(m, M, \boldsymbol{R}_D(h))$. Finally let $f : \triangle_M^m \to \mathbb{R}$ be defined by $f(x_1, \ldots, x_m) := \exp\big(\beta d\big(\frac{1}{m} \sum_{i=1}^m x_i, \boldsymbol{R}_D(h)\big)\big)$, which is convex since the average is linear, $d$ is convex and the exponential is non-decreasing and convex. Then, by swapping expectations (which is permitted by Fubini's theorem since the argument is non-negative) and applying Lemma 2, we have that $\mathbb{E}_{S' \sim D^m} Z_P(S')$ can be written as

$$
\begin{aligned}
\mathbb{E}_{S' \sim D^m} Z_P(S') &= \underset{S' \sim D^m}{\mathbb{E}} \ \underset{h \sim P}{\mathbb{E}} \exp\Big( \beta d\big(\boldsymbol{R}_{S'}(h), \boldsymbol{R}_D(h)\big) \Big) \\
&= \underset{h \sim P}{\mathbb{E}} \ \underset{S' \sim D^m}{\mathbb{E}} \exp\Big( \beta d\big(\boldsymbol{R}_{S'}(h), \boldsymbol{R}_D(h)\big) \Big) \\
&= \underset{h \sim P}{\mathbb{E}} \ \underset{\boldsymbol{X}_1, \ldots, \boldsymbol{X}_m}{\mathbb{E}} \exp\left( \beta d \left( \frac{1}{m} \sum_{i=1}^m \boldsymbol{X}_i, \boldsymbol{R}_D(h) \right) \right) \\
&\leq \underset{h \sim P}{\mathbb{E}} \ \underset{\boldsymbol{X}_1', \ldots, \boldsymbol{X}_m'}{\mathbb{E}} \exp\left( \beta d \left( \frac{1}{m} \sum_{i=1}^m \boldsymbol{X}_i', \boldsymbol{R}_D(h) \right) \right) \\
&= \underset{h \sim P}{\mathbb{E}} \ \underset{\boldsymbol{Y}}{\mathbb{E}} \exp\left( \beta d \left( \frac{1}{m} \boldsymbol{Y}, \boldsymbol{R}_D(h) \right) \right)
\end{aligned}
$$

---

[4]More precisely, when $\forall h \in \mathcal{H} \ \forall (x, y) \in S \ \exists j \in [M]$ such that $h(x)[\{\hat{y} \in \mathcal{Y} : (\hat{y}, y) \in E_j\}] = 1$.

$$= \mathop{\mathbb{E}}_{h \sim P} \sum_{\boldsymbol{k} \in S_{m,M}} \mathrm{Mult}\big(\boldsymbol{k}; m, M, \boldsymbol{R}_D(h)\big) \exp\left(\beta d\big(\tfrac{\boldsymbol{k}}{m}, \boldsymbol{R}_D(h)\big)\right)$$

$$\leq \sup_{\boldsymbol{r} \in \triangle_M} \left[\sum_{\boldsymbol{k} \in S_{m,M}} \mathrm{Mult}\big(\boldsymbol{k}; m, M, \boldsymbol{r}\big) \exp\left(\beta d\big(\tfrac{\boldsymbol{k}}{m}, \boldsymbol{r}\big)\right)\right],$$

which is the definition of $\mathcal{I}_d(m, \beta)$. Inequality (8) then follows by substituting this bound on $\mathbb{E}_{S' \sim D^m} Z_P(S')$ into (9) and dividing by $\beta$. $\qquad\square$

We now specialise Proposition 4 to the case $d(\cdot, \cdot) = \mathrm{kl}(\cdot\|\cdot)$ to obtain Corollary 1.

**Corollary 1.** *In the setting of Theorem 1,*

$$\mathrm{kl}\big(\boldsymbol{R}_S(Q)\|\boldsymbol{R}_D(Q)\big) \leq \frac{1}{m}\left[\mathrm{KL}(Q\|P) + \ln\frac{\eta(m, M)}{\delta}\right], \quad \text{where} \tag{10}$$

$$\eta(m, M) := \frac{m!}{m^m} \sum_{\boldsymbol{k} \in S_{m,M}} \prod_{j=1}^{M} \frac{k_j^{k_j}}{k_j!}. \tag{11}$$

*Proof.* Applying Proposition 4 with $d(\cdot, \cdot) = \mathrm{kl}(\cdot\|\cdot)$ and $\beta = m$ gives that with probability at least $1 - \delta$ over $S \sim D^m$, simultaneously for all posteriors $Q \in \mathcal{M}(\mathcal{H})$,

$$\mathrm{kl}\big(\boldsymbol{R}_S(Q)\|\boldsymbol{R}_D(Q)\big) \leq \frac{1}{m}\left[\mathrm{KL}(Q\|P) + \ln\frac{\mathcal{I}_{\mathrm{kl}}(m, m)}{\delta}\right],$$

where $\mathcal{I}_{\mathrm{kl}}(m, m) := \sup_{\boldsymbol{r} \in \triangle_M}[\sum_{\boldsymbol{k} \in S_{m,M}} \mathrm{Mult}(\boldsymbol{k}; m, M, \boldsymbol{r}) \exp\big(m\mathrm{kl}(\tfrac{\boldsymbol{k}}{m}, \boldsymbol{r})\big)]$. Thus it suffices to show that $\mathcal{I}_{\mathrm{kl}}(m, m) \leq \eta(m, M)$.

To prove this, for each fixed $\boldsymbol{r} = (r_1, \ldots, r_M) \in \triangle_M$ let $J_{\boldsymbol{r}} = \{j \in [M] : r_j = 0\}$. Then $\mathrm{Mult}(\boldsymbol{k}; m, M, \boldsymbol{r}) = 0$ for any $\boldsymbol{k} \in S_{m,M}$ such that $k_j \neq 0$ for some $j \in J_{\boldsymbol{r}}$. For the other $\boldsymbol{k} \in S_{m,M}$, namely those such that $k_j = 0$ for all $j \in J_{\boldsymbol{r}}$, the probability term can be written as $\mathrm{Mult}(\boldsymbol{k}; m, M, \boldsymbol{r}) = \frac{m!}{\prod_{j=1}^{M} k_j!} \prod_{j=1}^{M} r_j^{k_j} = \frac{m!}{\prod_{j \notin J_{\boldsymbol{r}}} k_j!} \prod_{j \notin J_{\boldsymbol{r}}} r_j^{k_j}$, and (recalling the convention that $0 \ln\frac{0}{0} = 0$) the term $\exp(m\mathrm{kl}(\tfrac{\boldsymbol{k}}{m}, \boldsymbol{r}))$ can be written as

$$\exp\left(m \sum_{j=1}^{M} \tfrac{k_j}{m} \ln\frac{\tfrac{k_j}{m}}{r_j}\right) = \exp\left(\sum_{j \notin J_{\boldsymbol{r}}} k_j \ln\frac{k_j}{mr_j}\right) = \prod_{j \notin J_{\boldsymbol{r}}} \left(\frac{k_j}{mr_j}\right)^{k_j} = \frac{1}{m^m} \prod_{j \notin J_{\boldsymbol{r}}} \left(\frac{k_j}{r_j}\right)^{k_j},$$

where the last equality is obtained by recalling that the $k_j$ sum to $m$. Substituting these two expressions into the definition of $\mathcal{I}_{\mathrm{kl}}(m, m)$ and only summing over those $\boldsymbol{k} \in S_{m,M}$ with non-zero probability, we obtain

$$\sum_{\boldsymbol{k} \in S_{m,M}} \mathrm{Mult}(\boldsymbol{k}; m, M, \boldsymbol{r}) \exp\big(m\mathrm{kl}\big(\tfrac{\boldsymbol{k}}{m}, \boldsymbol{r}\big)\big) = \sum_{\substack{\boldsymbol{k} \in S_{m,M}: \\ \forall j \in J_{\boldsymbol{r}} \; k_j = 0}} \mathrm{Mult}(\boldsymbol{k}; m, M, \boldsymbol{r}) \exp\big(m\mathrm{kl}\big(\tfrac{\boldsymbol{k}}{m}, \boldsymbol{r}\big)\big)$$

$$= \sum_{\substack{\boldsymbol{k} \in S_{m,M}: \\ \forall j \in J_{\boldsymbol{r}} \; k_j = 0}} \frac{m!}{\prod_{j \notin J_{\boldsymbol{r}}} k_j!} \prod_{j \notin J_{\boldsymbol{r}}} r_j^{k_j} \frac{1}{m^m} \prod_{j \notin J_{\boldsymbol{r}}} \left(\frac{k_j}{r_j}\right)^{k_j}$$

$$= \frac{m!}{m^m} \sum_{\substack{\boldsymbol{k} \in S_{m,M}: \\ \forall j \in J_{\boldsymbol{r}} \; k_j = 0}} \prod_{j \notin J_{\boldsymbol{r}}} \frac{k_j^{k_j}}{k_j!}$$

$$= \frac{m!}{m^m} \sum_{\substack{\boldsymbol{k} \in S_{m,M}: \\ \forall j \in J_{\boldsymbol{r}} \; k_j = 0}} \prod_{j=1}^{M} \frac{k_j^{k_j}}{k_j!} \qquad \text{(because } \tfrac{0^0}{0!} = 1\text{)}$$

$$\leq \frac{m!}{m^m} \sum_{\boldsymbol{k} \in S_{m,M}} \prod_{j=1}^{M} \frac{k_j^{k_j}}{k_j!},$$

which is $\eta(m, M)$. Since this is independent of $\boldsymbol{r}$, it also holds after taking the supremum over $\boldsymbol{r} \in \triangle_M$ of the left hand side, showing that $\mathcal{I}_{\mathrm{kl}}(m, m) \leq \eta(m, M)$. $\qquad\square$

The final step in obtaining Theorem 1 is to loosen the bound given in Corollary 1 (which is intractable when $m$ is large) to the tractable form given in Theorem 1. For this we require the following technical lemma, the proof of which we defer to Appendix C.3.

**Lemma 3.** *For integers $M \geq 1$ and $m \geq M$, $\sum_{\boldsymbol{k} \in S_{m,M}^{>0}} \frac{1}{\prod_{j=1}^M \sqrt{k_j}} \leq \frac{\pi^{\frac{M}{2}} m^{\frac{M-2}{2}}}{\Gamma(\frac{M}{2})}$.*

*Proof.* (Of Theorem 1) It suffices to show that for all $m \geq M \geq 1$ we have $\eta(m, M) \leq \xi(m, M)$. We achieve this by applying Stirling's approximation $\sqrt{2\pi n} \left(\frac{n}{e}\right)^n < n! < \sqrt{2\pi n} \left(\frac{n}{e}\right)^n e^{\frac{1}{12n}}$ (valid for $n \geq 1$) to the factorials in $\eta(m, M)$ and then using Lemma 3.

Since Stirling's approximation requires that all the $k_j$ are at least one, we partition the sum in $\eta(m, M)$ according to the number of coordinates of $\boldsymbol{k}$ at which $k_j = 0$. Let $z$ index the number of such coordinates. Defining $f : \bigcup_{M=2}^\infty S_{m,M} \to \mathbb{R}$ by $f(\boldsymbol{k}) = \prod_{j=1}^{|\boldsymbol{k}|} k_j^{k_j}/k_j!$ and noting that $f$ is symmetric under permutations of its arguments, we then have

$$\eta(m, M) = \frac{m!}{m^m} \sum_{\boldsymbol{k} \in S_{m,M}} f(\boldsymbol{k}) = \frac{m!}{m^m} \sum_{z=0}^{M-1} \binom{M}{z} \sum_{\boldsymbol{k} \in S_{m,M-z}^{>0}} f(\boldsymbol{k}). \tag{12}$$

Stirling's approximation can now be applied to each $\boldsymbol{k} \in S_{m,M}^{>0}$ $f(\boldsymbol{k}) \leq \prod_{j=1}^M \frac{k_j^{k_j}}{\sqrt{2\pi k_j}\left(\frac{k_j}{e}\right)^{k_j}} =$

$\prod_{j=1}^M \frac{e^{k_j}}{\sqrt{2\pi k_j}} = \frac{e^m}{(2\pi)^{M/2}} \prod_{j=1}^M \frac{1}{\sqrt{k_j}}$. An application of Lemma 3 now gives

$$\sum_{\boldsymbol{k} \in S_{m,M-z}^{>0}} f(\boldsymbol{k}) \leq \sum_{\boldsymbol{k} \in S_{m,M-z}^{>0}} \frac{e^m}{(2\pi)^{\frac{M-z}{2}}} \prod_{j=1}^{M-z} \frac{1}{\sqrt{k_j}} \leq \frac{e^m}{(2\pi)^{\frac{M-z}{2}}} \frac{\pi^{\frac{M-z}{2}} m^{\frac{M-z-2}{2}}}{\Gamma\left(\frac{M-z}{2}\right)} = \frac{e^m m^{\frac{M-z-2}{2}}}{2^{\frac{M-z}{2}} \Gamma\left(\frac{M-z}{2}\right)}.$$

Substituting this into equation (12) and bounding $m!$ using Stirling's approximation, we have $\eta(m, M) \leq \frac{\sqrt{2\pi m} e^{1/(12m)}}{e^m} \sum_{z=0}^{M-1} \binom{M}{z} \frac{e^m m^{\frac{M-z-2}{2}}}{2^{\frac{M-z}{2}} \Gamma\left(\frac{M-z}{2}\right)} = \xi(m, M)$, which completes the proof of the bound. As for the order of the bound, it is sufficient to bound $\ln \xi(m, M)$ using the crude approximations $\binom{M}{z} \leq M^M$, $(2/m)^{z/2} \leq 1$ and $\Gamma((M-z)/2) \geq 1$.

$\qquad\square$

## 6 Numerical experiments

We use binarised versions of MNIST, and HAM10000 Tschandl [2018]. In both cases we partition $\mathcal{Y}^2$ into $E_0 = \{(0,0), (1,1)\}$, $E_1 = \{(0,1)\}$ and $E_2 = \{(1,0)\}$, and take $\boldsymbol{\ell} = (0, 1, 3)$. Each dataset is split into prior and certification sets. We take $\mathcal{H}$ to be two-layer MLPs. As is common in the PAC-Bayes literature, we restrict $P$ and $Q$ to be isotropic and diagonal Gaussian distributions over the parameter space, respectively. The means of $P$ and $Q$ are set to the parameters of an MLP trained on the prior set. The mean and variances of $Q$ and the variance of $P$ are tuned via Theorem 2 to minimize the bound on the total risk $R_D^T(Q)$. See Appendix A for pseudocode, Appendix B for full experimental details and https://github.com/reubenadams/PAC-Bayes-Control for code. The results for MNIST can be seen in Figure 1.

We estimate $\boldsymbol{R}_S(Q)$ with a Monte Carlo and obtain a PAC-Bayes bound on $R_D^T(Q)$ by combining Proposition 2 (with $\delta = 0.01$ and $N = 100000$) and Proposition 3. We obtain $R_D^T(Q) \leq 0.2640$ for MNIST and $R_D^T(Q) \leq 0.8379$ for HAM10000, where both bounds hold with probability at least $1 - 0.05 - 0.01 = 0.94$. While these bounds are far from vacuous—the maximum possible value of $R_D^T(Q)$ is 3 for our choice of $\boldsymbol{\ell}$—one might wonder whether one can do better by bounding each error probability individually using Maurer's inequality Maurer [2004], and then unioning these bounds. As with our Theorem 1, this would also constrain the entire distribution of error types since for any $\boldsymbol{\ell}$, one could then calculate the maximimum value of $R_D^T(Q)$ that satisfies all of these constraints. Both

| Dataset | Volume Our Region | Volume Maurer Region |
|---------|-------------------|----------------------|
| MNIST | **0.0025** (0.002498, 0.002504) | 0.0028 (0.002793, 0.002800) |
| HAM10000 | 0.0012 (0.001207, 0.001211) | **0.0011** (0.001142, 0.001146) |

Table 1: Point estimates and 95% confidence intervals for the volumes of the confidence regions for $\boldsymbol{R}_D(Q)$ given by Theorem 1 and a union over $M$ individual Maurer bounds, respectively. Our method is superior for MNIST and inferior for HAM100000.

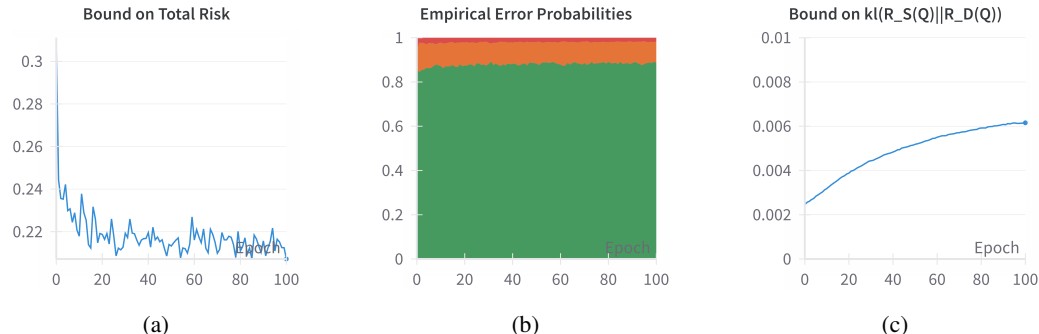

(a)            (b)            (c)

Figure 1: Experimental results for binarised MNIST. (a) The PAC-Bayes bound on the total risk decreases when tuning the posterior via Theorem 2. (b) This is achieved by a shift in the empirical error probabilities. (c) The bound on $\mathrm{kl}(\boldsymbol{R}_S(Q)\|\boldsymbol{R}_D(Q))$ is not substantially increased, meaning we still retain good control of $\boldsymbol{R}_D(Q)$ after optimizing $Q$ for this particular choice of $\ell$.

methods constrain the region of the simplex in which $\boldsymbol{R}_D(Q)$ can lie (with high probability), and a reasonable metric by which to compare them is the volumes of these regions. This can be estimated via a MC sample by uniformly sampling points $\boldsymbol{r}$ from $\triangle_M$ and counting how samples are legal values of $\boldsymbol{R}_D(Q)$ according to each method. The 95% confidence intervals for the volumes of the two regions are given in Table 1. A more comprehensive table for synthetic values of $\boldsymbol{R}_S(Q)$ can be found in Appendix B.

# 7 Perspectives

We introduce the framework of error types, considering the vectors $\boldsymbol{R}_S(Q)$ and $\boldsymbol{R}_D(Q)$ of empirical and true probabilities of errors of different types. We prove a PAC-Bayes bound (Theorem 1) on $\mathrm{kl}(\boldsymbol{R}_S(Q)\|\boldsymbol{R}_D(Q))$ which controls the entire distribution of error probabilities, and hence can be used to derive bounds on arbitrary linear combinations of the error probabilities, all of which hold simultaneously with high probability; this cannot be achieved with any existing PAC-Bayes bound.

We construct a differential training objective based on our bound by introducing the the vectorised kl inverse, providing a recipe for quickly computing its value and derivatives (Theorem 2). Our framework is flexible enough to encompass multiclass classification or discretised regression, but also structured output prediction, multi-task learning and learning-to-learn.

Another potential application of our work is to the excess risk, since under a misclassification loss there are three different error types, corresponding to excess losses of $\{-1, 0, 1\}$. Biggs and Guedj [2023] adapted Theorems 1 and 2 to this setting, leading to an empirically tighter PAC-Bayes bound for certain classification tasks.

We require i.i.d. data, which in practice is frequently not the case or is hard to verify. Further, the number of error types $M$ must be finite. In continuous scenarios it would be preferable to be able to control the entire distribution of loss values without having to discretise into finitely many error types. We leave this direction to future work.

## Acknowledgments and Disclosure of Funding

We warmly thank reviewers and the Area Chair who provided insigthful comments and suggestions which greatly helped us improve our manuscript. R.A. was supported by the UKRI grant number EP/S021566/1 and gratefully thanks Felix Biggs for his insights. J.S-T gratefully acknowledges the European Union's Horizon 2020 Research and Innovation Program through the grant numbers 951847 (European learning and intelligent systems excellence, ELISE) and 952026 (Human-centred artificial intelligence, HumanE-AI-Net). B.G. acknowledges partial support by the U.S. Army Research Laboratory and the U.S. Army Research Office, and by the U.K. Ministry of Defence and the U.K. Engineering and Physical Sciences Research Council (EPSRC) under grant number EP/R013616/1. B.G. acknowledges partial support from the French National Agency for Research, through grants ANR-18-CE40-0016-01 and ANR-18- CE23-0015-02, and through the programme "France 2030" and PEPR IA on grant SHARP ANR-23-PEIA-0008.

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

# A  Recipe for implementing Theorems 1 and 2

We here outline more explicitly how Theorem 1 and Theorem 2 may be used to formulate a fully differentiable objective by which a model may be trained.

First, if one wishes to make hard labels, namely $\mathcal{H} \subseteq \mathcal{Y}^{\mathcal{X}}$, it will first be necessary to use a surrogate class of soft hypotheses $\mathcal{H}' \subseteq \mathcal{M}(\mathcal{Y})^{\mathcal{X}}$ during training, before reverting to hard labels for example by taking the mean label or the one with highest probability. Using soft hypotheses during training is necessary to ensure that the empirical $j$-risks $R_S^j(Q)$ are differentiable with respect to the model parameters. Since how one chooses to do this will depend on the specific use case, we restrict our attention here to the case of soft hypotheses. Specifically, we consider a class of soft hypotheses $\mathcal{H} = \{h_\theta : \theta \in \mathbb{R}^N\} \subseteq \mathcal{M}(\mathcal{Y})^{\mathcal{X}}$ parameterised by the weights $\theta \in \mathbb{R}^N$ of some neural network of a given architecture with $N$ parameters in such a way that the $R_S^j(h_\theta)$ are differentiable in $\theta$. A concrete example would be multiclass classification using a fully connected neural network with output being softmax probabilities on the classes so that the $R_S^j(h_\theta)$ are differentiable.

Second, it is necessary to restrict the prior and posterior $P, Q \in \mathcal{M}(\mathcal{H})$ to a parameterised subset of $\mathcal{M}(\mathcal{H})$ in which $\mathrm{KL}(Q\|P)$ has a closed form which is differentiable in the parameterisation. A simple choice for our case of a neural network with $N$ parameters is $P, Q \in \{\mathcal{N}(\boldsymbol{w}, \mathrm{diag}(\boldsymbol{s})) : \boldsymbol{w} \in \mathbb{R}^N, \boldsymbol{s} \in \mathbb{R}_{>0}^N\}$. For prior a $P_{\boldsymbol{v},\boldsymbol{r}} = \mathcal{N}(\boldsymbol{v}, \mathrm{diag}(\boldsymbol{r}))$ and posterior $Q_{\boldsymbol{w},\boldsymbol{s}} = \mathcal{N}(\boldsymbol{w}, \mathrm{diag}(\boldsymbol{s}))$ we have the closed form

$$\mathrm{KL}(Q_{\boldsymbol{w},\boldsymbol{s}}\|P_{\boldsymbol{v},\boldsymbol{r}}) = \frac{1}{2}\left[\sum_{n=1}^N \left(\frac{s_n}{r_n} + \frac{(w_n - v_n)^2}{r_n} + \ln\frac{r_n}{s_n}\right) - N\right],$$

which is indeed differentiable in $\boldsymbol{v}, \boldsymbol{r}, \boldsymbol{w}$ and $\boldsymbol{s}$. While $Q_{\boldsymbol{w},\boldsymbol{s}}$ and $P_{\boldsymbol{v},\boldsymbol{r}}$ are technically distributions on $\mathbb{R}^D$ rather than $\mathcal{H}$, the KL-divergence between the distributions they induce on $\mathcal{H}$ will be at most as large as the expression above. Thus, substituting the expression above into the bounds we prove in Section 3 can only increase the value of the bounds, meaning the enlarged bounds certainly still hold with probability at least $1 - \delta$.

Third, in all but the simplest cases $R_S^j(Q_{\boldsymbol{w},\boldsymbol{s}})$ will not have a closed form, much less one that is differentiable in $\boldsymbol{w}$ and $\boldsymbol{s}$. A common solution to this is to use the so-called pathwise gradient estimator. In our case, this corresponds to drawing $\boldsymbol{\epsilon} \sim \mathcal{N}(\boldsymbol{0}, \mathbb{I})$, where $\mathbb{I}$ is the $N \times N$ identity matrix, and estimating

$$\nabla_{\boldsymbol{w},\boldsymbol{s}} R_S^j(Q_{\boldsymbol{w},\boldsymbol{s}}) = \nabla_{\boldsymbol{w},\boldsymbol{s}}\left[\mathbb{E}_{\boldsymbol{\epsilon}' \sim \mathcal{N}(\boldsymbol{0}, \mathbb{I})} R_S^j(h_{\boldsymbol{w}+\boldsymbol{\epsilon}' \odot \sqrt{\boldsymbol{s}}})\right] \approx \nabla_{\boldsymbol{w},\boldsymbol{s}} R_S^j(h_{\boldsymbol{w}+\boldsymbol{\epsilon} \odot \sqrt{\boldsymbol{s}}}),$$

where $h_{\boldsymbol{w}}$ denotes the function expressed by the neural network with parameters $\boldsymbol{w}$. For a proof that this is an unbiased estimator, and for other methods for estimating the gradients of expectations, see the survey Mohamed et al. [2020].

Fourth, one must choose the prior. Designing priors which are optimal in some sense (*i.e.*, minimising the Kullback-Leibler term in the right-hand side of generalisation bounds) has been at the core of an active line of work in the PAC-Bayesian literature. For the sake of simplicity, and since it is out of the scope of our contributions, we assume here that the prior is given beforehand, although we stress that practitioners should pay great attention to its tuning. For our purposes, it suffices to say that if one is using a data-dependent prior then it is necessary to partition the sample into $S = S_{\mathrm{Prior}} \cup S_{\mathrm{Bound}}$, where $S_{\mathrm{Prior}}$ is used to train the prior and $S_{\mathrm{Bound}}$ is used to evaluate the bound. Since our bound holds uniformly over posteriors $Q \in \mathcal{M}(\mathcal{H})$, the entire sample $S$ is free to be used to train the posterior $Q$. For a more in-depth discussion on the choice of prior, we refer to the following body of work: Ambroladze et al. [2006], Lever et al. [2010, 2013], Parrado-Hernández et al. [2012], Dziugaite and Roy [2017, 2018], Rivasplata et al. [2018], Letarte et al. [2019], Pérez-Ortiz et al. [2021], Dziugaite et al. [2021], Biggs and Guedj [2022a,b].

Finally, given a confidence level $\delta \in (0, 1]$, one may use Algorithm 1 to obtain a posterior $Q_{\boldsymbol{w},\boldsymbol{s}}$ with minimal upper bound on the total risk. Note we take the pointwise logarithm of the variances $\boldsymbol{r}$ and $\boldsymbol{s}$ to obtain unbounded parameters on which to perform stochastic gradient descent or some other minimisation algorithm. We use $\oplus$ to denote vector concatenation. The algorithm can be straightforwardly adapted to permit mini-batches by, for each epoch, sequentially repeating the steps with $S$ equal to each mini-batch.

**Input:**
$\mathcal{X}, \mathcal{Y}$ /* Arbitrary input and output spaces                                       */
$\bigcup_{j=1}^{M} E_j = \mathcal{Y}^2$ /* A finite partition into error types                    */
$\boldsymbol{\ell} \in [0, \infty)^M$ /* A vector of losses, not all equal                          */
$S = S_{\text{Prior}} \cup S_{\text{Bound}} \in (\mathcal{X} \times \mathcal{Y})^m$ /* A partitioned i.i.d. sample  */
$N \in \mathbb{N}$ /* The number of model parameters                                          */
$P_{\boldsymbol{v}, \boldsymbol{r}}, \, \boldsymbol{v}(S_{\text{Prior}}) \in \mathbb{R}^N, \boldsymbol{r}(S_{\text{Prior}}) \in \mathbb{R}_{\geq 0}^N$ /* A (data-dependent) prior  */
$Q_{\boldsymbol{w}_0, \boldsymbol{s}_0}, \, \boldsymbol{w}_0 \in \mathbb{R}^N, \boldsymbol{s}_0 \in \mathbb{R}_{\geq 0}^N$ /* An initial posterior             */
$\delta \in (0, 1]$ /* A confidence level                                                    */
$\lambda > 0$ /* A learning rate                                                           */
$T$ /* The number of epochs to train for                                            */

**Output:**
$Q_{\boldsymbol{w}, \boldsymbol{s}}, \, \boldsymbol{w} \in \mathbb{R}^N, \boldsymbol{s} \in \mathbb{R}_{\geq 0}^N$ /* A trained posterior                      */

**Procedure:**
$\boldsymbol{\zeta}_0 \leftarrow \log \boldsymbol{s}_0$ /* Transform to unbounded scale parameters           */
$\boldsymbol{p} \leftarrow \boldsymbol{w}_0 \oplus \boldsymbol{\zeta}_0$ /* Collect mean and scale parameters                  */
**for** $t \leftarrow 1$ **to** $T$ **do**
    Draw $\boldsymbol{\epsilon} \sim \mathcal{N}(\boldsymbol{0}, \mathbb{I})$
    $\boldsymbol{u} \leftarrow \boldsymbol{R}_S \left( h_{\boldsymbol{w} + \boldsymbol{\epsilon} \odot \sqrt{\exp(\boldsymbol{\zeta})}} \right)$
    $B \leftarrow$
    $\frac{1}{m} \left[ \text{KL} \left( Q_{\boldsymbol{w}, \exp(\boldsymbol{\zeta})} \big\| P_{\boldsymbol{v}, \boldsymbol{r}} \right) + \ln \left( \frac{1}{\delta} \sqrt{\pi} e^{1/12m} \left( \frac{m}{2} \right)^{\frac{M-1}{2}} \sum_{z=0}^{M-1} \binom{M}{z} \frac{1}{(\pi m)^{z/2} \Gamma\left( \frac{M-z}{2} \right)} \right) \right]$

    $\tilde{\boldsymbol{u}} \leftarrow (u_1, \ldots, u_M, B)$
    $\boldsymbol{G} \leftarrow \boldsymbol{0}_{2N \times (M+1)}$ /* Initialise gradient matrix                        */
    $\boldsymbol{F} \leftarrow \boldsymbol{0}_{M+1}$ /* Initialise gradient vector                         */
    **for** $j \leftarrow 1$ **to** $M + 1$ **do**
        $\boldsymbol{F}_j \leftarrow \frac{\partial f_{\boldsymbol{\ell}}^*}{\partial \tilde{u}_j}(\tilde{\boldsymbol{u}})$ /* Gradients of total loss from Theorem 2      */
        **for** $i \leftarrow 1$ **to** $2N$ **do**
            $\boldsymbol{G}_{i,j} \leftarrow \frac{\partial \tilde{u}_j}{\partial p_i}(\boldsymbol{p})$ /* Gradients of empirical risks and bound     */
        **end**
    **end**
    $\boldsymbol{H} \leftarrow \boldsymbol{G}\boldsymbol{F}$ /* Gradients of total loss w.r.t. parameters         */
    $\boldsymbol{p} \leftarrow \boldsymbol{p} - \lambda \boldsymbol{H}$ /* Gradient step                                    */
**end**
$\boldsymbol{w} = (p_1, \ldots, p_N)$
$\boldsymbol{s} = (\exp(p_{N+1}), \ldots, \exp(p_{2N}))$
**return** $\boldsymbol{w}, \boldsymbol{s}$

**Algorithm 1:** Calculating a posterior with minimal bound on the total risk.

# B   Additional Experimental Details

For MNIST we map labels $\{0, 1, 2, 3, 4\}$ to 0 and $\{5, 6, 7, 8, 9\}$ to 1. For HAM10000 we map the cancerous or pre-cancerous labels $\{$`Melanoma`, `Basal Cell Carcinoma`, `Actinic Keratosis`$\}$ to 1 and the other labels to 0. In both cases we partition $\mathcal{Y}^2$ into $E_0 = \{(0,0), (1,1)\}$, $E_1 = \{(0,1)\}$ and $E_2 = \{(1,0)\}$, and take $\boldsymbol{\ell} = (0, 1, 3)$. For HAM10000, $E_1$ and $E_2$ then refer to Type I and Type II errors, respectively, and $\boldsymbol{\ell}$ reflects the greater severity of false negatives.

Each dataset is split into prior and certification sets $S_{\text{Prior}}$ and $S_{\text{Bound}}$, respectively. For MNIST, we use the conventional training set of size 60000 as the prior set, and the conventional test set of size 10000 as the certification set. For HAM10000 we pool the conventional train, validation and test sets together and then split 50-50 to obtain prior and certification sets each of size 5860. For HAM10000

we resize the images to $(28, 28)$ and use just the first channel so that the data dimension is the same for both datasets.

We take $\mathcal{H}$ to be two-layer MLPs with 784, 100 and 2 units in the input, hidden and output layers, respectively. As is common in the PAC-Bayes literature, we restrict $P$ to be an isotropic Gaussian $N(\boldsymbol{v}, \lambda\boldsymbol{I})$ and $Q$ to be a diagonal Gaussian $N(\boldsymbol{w}, \operatorname{diag}(\boldsymbol{s}))$. Further, as in Dziugaite and Roy [2017], we restrict $\lambda$ to be of the form $\lambda_j = c\exp(-j/b)$ for some $j \in \mathbb{N}$, taking $c = 0.1$ and $b = 100$. Since, at the end of training, we will then have one prior $P_j$ for each $j \in \mathbb{N}$, we can choose the $j$ that minimizes the PAC-Bayes bound provided we take a union over all of them, taking $\delta_j = \frac{6\delta}{\pi^2 j^2}$ so that $\sum_j \delta_j = 1$ and all the bounds hold simultaneously with probability at least $1 - \delta$. After applying Algorithm 1 we round $\lambda$ to a discrete $\lambda_j$, either up or down depending on which gives the smaller bound.

For both datasets we set the prior mean $\boldsymbol{v}$ to be the parameters of an MLP trained on the prior set. In both cases we use SGD with learning rate 0.01 to minimise the cross-entropy loss, using a portion of the prior set as a validation set. For MNIST we train the MLP for 20 epochs to get an error rate of 14%, for HAM10000 we train the MLP for 5 epochs to get an error rate of 22%. We then apply Algorithm 1. By combining Proposition 2 (with $\delta = 0.01$ and $N = 100000$) and Proposition 3. We obtain $\boldsymbol{R}_S(\hat{Q}) = (0.8879, 0.0919, 0.0203)$ and $R_D^T(Q) \le 0.2640$ for MNIST and $\boldsymbol{R}_S(\hat{Q}) = (0.7860, 0.0146, 0.1995)$ and $R_D^T(Q) \le 0.8379$ for HAM10000, where both bounds hold with probability at least $1 - 0.05 - 0.01 = 0.94$.

The full results are shown in Figure 2. Figures 2a, 2c and 2e are the same as Figures 1a, 1b and 1c, and are repeated here for easier comparison with the HAM10000 results. Figure 2b shows that Algorithm 1 has failed to reduce the bound on the total risk beyond the initialisation of $Q$ to $P$, with the small variation being explained by different MC samples being drawn from $Q$ during training rather than $Q$ changing substantially. Indeed, Figure 2h shows that $Q$ does not appreciably move from its initialisation at $P$—$\mathrm{KL}(Q\|P)$ remains below 0.1 whereas in the MNNIST experiment, which has the same number of parameters, exceeds 30. It is therefore unsurprising that Figures 2d and 2f show negligible change in the empirical error probabilities and the bound on $\mathrm{kl}(\boldsymbol{R}_S(Q)\|\boldsymbol{R}_D(Q))$, respectively. The divergence in the results is likely due to the difference in sample size; the certification set for the MNIST experiment contains 10000 samples, whereas for the HAM10000 dataset there are only 5000, which, all else equal, makes an increase in $\mathrm{KL}(Q\|P)$ twice as expensive.

Recall from Section 6 that while $\boldsymbol{R}_D(Q)$ can be effectively constrained to a sub-region of the simple $\triangle_M$ using our Theorem 1, this can also be achieved by unioning $M$ Maurer bounds, one for each error probability. Table 1 gave the 95% confidence intervals for the volumes of the confidence regions in which $\boldsymbol{R}_D(Q)$ was likely to lie for experiments on MNIST and HAM10000, but neither region was uniformly smaller, making it unclear which method should be preferred.

Table 2 provides additional data by taking synthetic values for $\boldsymbol{R}_S(Q)$ and $\mathrm{KL}(Q\|P)$, for different values of $m$ (the size of the certification set) and $M$ (the number of error types). 'Individual' denotes unioning individual Maurer bounds, 'Ours' is our method, 'Intersection' is the intersection of the confidence regions given by the previous two methods (but loosened so that they now both hold simultaneously with probability at least 0.95), and 'Morv.' is the confidence region produced by Morvant's bound Morvant et al. [2012]. The 95% confidence intervals for the volumes of all the regions have been produced by Monte Carlo samples. We see that our confidence region is tighter than the individual one in 4/9 cases (green), worse in 3/9 cases (red) and ties in 2/9 cases (orange). Interestingly, union bounding the naive CR and our CR and intersecting often beats both of these (**bold**). Morvant's result is either not applicable or their confidence region is much larger than ours and essentially takes up the entire simplex, hence the volume estimate of 1.000. The reason their bound is sometimes inapplicable is because it requires every class to contain at least $8L$ instances, where $L$ is the number of labels—in the $L = 5$, $M = 25$, $m = 100$ case this would require each class to contain at least $5 \times 8 = 40$ instances which is impossible with $m = 100$ samples.

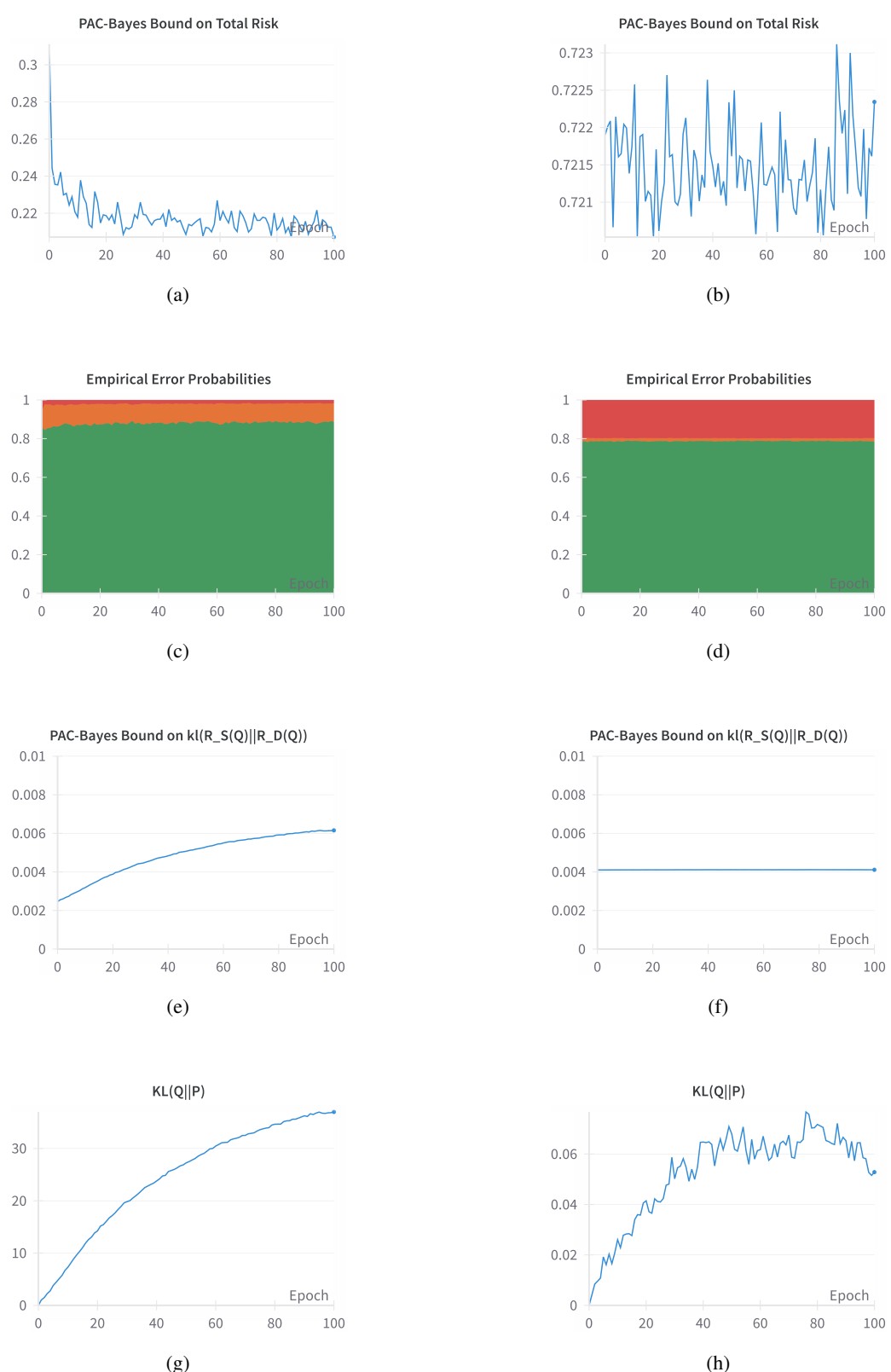

Figure 2: MNIST (first column) and HAM10000 (second column) experiments.

| $M$ | $m$ | Vol. Individual | Vol. Ours | Vol. Intersection | Vol. Morv. |
|---|---|---|---|---|---|
| | 100 | (0.1195, 0.1196) | (0.1165, 0.1166) | **(0.1160, 0.1161)** | (1.0, 1.0) |
| $2^2$ | 300 | (0.02920, 0.02926) | (0.03071, 0.03078) | **(0.02893, 0.02900)** | (1.0, 1.0) |
| | 1000 | (5.635e-3, 5.664e-3) | (6.475e-3, 6.507e-3) | (5.706e-3, 5.735e-3) | (1.0, 1.0) |
| | 100 | (0.3190, 0.3192) | (0.1757, 0.1758) | **(0.1582, 0.1584)** | N/A |
| $5^2$ | 300 | (1.306e-3, 1.320e-3) | (3.672e-4, 3.748e-4) | **(2.515e-4, 2.578e-4)** | (1.0, 1.0) |
| | 1000 | (1.090e-08, 1.024-07) | (2.422e-09, 7.225e-08) | (0.000, 3.689e-08) | (1.0, 1.0) |
| | 100 | (0.9990, 0.9990) | (1.000, 1.000) | (0.9995, 0.9995) | N/A |
| $10^2$ | 300 | (0.3534, 0.3536) | (0.1688, 0.1689) | **(0.1306, 0.1307)** | N/A |
| | 1000 | (3.454e-8, 1.5763e-7) | (0.000, 3.688e-8) | (0.000, 3.688e-8) | (1.0, 1.0) |

Table 2: 95% confidence intervals for the volumes of the confidence regions for $\boldsymbol{R}_D(Q)$. We set $\mathrm{KL}(Q\|P) = 0$, $\delta = 0.05$, $\boldsymbol{R}_S(Q) = (1/M, ..., 1/M)$ and use $10^8$ Monte Carlo samples.

## C   Proofs

### C.1   Proof of Proposition 3

Write $\mathrm{kl}(\hat{\boldsymbol{q}}\|\boldsymbol{p})$ as

$$\sum_{j=1}^{M} \hat{q}_j \ln \frac{\hat{q}_j}{q_j} + \sum_{j=1}^{M} \hat{q}_j \ln \frac{q_j}{p_j}. \tag{13}$$

The result then follows by bounding the two sums by

$$\sum_{j=1}^{M} \hat{q}_j \ln \frac{\hat{q}_j}{q_j} = \sum_{j=1}^{M} \mathrm{kl}(\hat{q}_j \| q_j) - (1 - \hat{q}_j) \ln \frac{1 - \hat{q}_j}{1 - q_j} \leq MB_2 - \sum_{j=1}^{M} (1 - \hat{q}_j) \ln \frac{1 - \hat{q}_j}{1 - \underline{q}_j} \tag{14}$$

and

$$\sum_{j=1}^{M} \hat{q}_j \ln \frac{q_j}{p_j} = \sum_{j=1}^{M} \frac{\hat{q}_j}{q_j} q_j \ln \frac{q_j}{p_j} \leq \max_{j} \frac{\hat{q}_j}{\underline{q}_j} \sum_{j=1}^{M} q_j \ln \frac{q_j}{p_j} \leq B_1 \max_{j} \frac{\hat{q}_j}{\underline{q}_j}. \tag{15}$$

Putting these together we obtain the bound on $\mathrm{kl}(\hat{\boldsymbol{q}}\|\boldsymbol{p})$. The limit follows because each $\underline{q}_j \to \hat{q}_j$ as $B_2 \to 0$.

### C.2   Proof of Lemma 2

Let $\boldsymbol{E}_M := \{\boldsymbol{e}_1, \ldots, \boldsymbol{e}_M\}$, namely the set of $M$-dimensional basis vectors. We will denote a typical element of $\boldsymbol{E}_M^m$ by $\boldsymbol{\eta}^{(m)} = (\boldsymbol{\eta}_1, \ldots, \boldsymbol{\eta}_m)$. For any $\boldsymbol{x}^{(m)} = (\boldsymbol{x}_1, \ldots, \boldsymbol{x}_m) \in \triangle_M^m$, a straightforward induction on $m$ yields

$$\sum_{\boldsymbol{\eta}^{(m)} \in \boldsymbol{E}_M^m} \left( \prod_{i=1}^{m} \boldsymbol{x}_i \cdot \boldsymbol{\eta}_i \right) = 1. \tag{16}$$

To see this, for $m = 1$ we have $\boldsymbol{E}_M^1 = \{(\boldsymbol{e}_1, ), \ldots, (\boldsymbol{e}_M, )\}$, where we have been pedantic in using 1-tuples to maintain consistency with larger values of $m$. Thus, for any $\boldsymbol{x}^{(1)} = (\boldsymbol{x}_1, ) \in \triangle_M^1$, the left hand side of equation (16) can be written as

$$\sum_{j=1}^{M} \boldsymbol{x}_1 \cdot \boldsymbol{e}_j = \sum_{j=1}^{M} (\boldsymbol{x}_1)_j = 1.$$

Now suppose that equation (16) holds for any $\boldsymbol{x}^{(m)} \in \triangle_M^m$ and let $\boldsymbol{x}^{(m+1)} = (\boldsymbol{x}_1, \ldots, \boldsymbol{x}_{m+1}) \in \triangle_M^{m+1}$. Then the left hand side of equation (16) can be written as

$$\sum_{\boldsymbol{\eta}^{(m+1)} \in \boldsymbol{E}_M^{m+1}} \left( \prod_{i=1}^{m+1} \boldsymbol{x}_i \cdot \boldsymbol{\eta}_i \right) = \sum_{\boldsymbol{\eta}^{(m)} \in \boldsymbol{E}_M^m} \sum_{j=1}^{M} \left( \prod_{i=1}^{m} \boldsymbol{x}_i \cdot \boldsymbol{\eta}_i \right) (\boldsymbol{x}_{m+1} \cdot \boldsymbol{e}_j)$$

$$= \sum_{\boldsymbol{\eta}^{(m)} \in \boldsymbol{E}_M^m} \left( \prod_{i=1}^{m} \boldsymbol{x}_i \cdot \boldsymbol{\eta}_i \right) \sum_{j=1}^{M} (\boldsymbol{x}_{m+1} \cdot \boldsymbol{e}_j) = 1.$$

We now show that any $\boldsymbol{x}^{(m)} = (\boldsymbol{x}_1, \ldots, \boldsymbol{x}_m) \in \triangle_M^m$ can be written as a convex combination of the elements of $\boldsymbol{E}_M^m$ in the following way

$$\boldsymbol{x}^{(m)} = \sum_{\boldsymbol{\eta}^{(m)} \in \boldsymbol{E}_M^m} \left( \prod_{i=1}^{m} \boldsymbol{x}_i \cdot \boldsymbol{\eta}_i \right) \boldsymbol{\eta}^{(m)}. \tag{17}$$

We have already shown that the weights sum to one, and they are clearly elements of $[0, 1]$, so the right hand side of equation (17) is indeed a convex combination of the elements of $\boldsymbol{E}_M^m$. We now show that equation (17) holds, again by induction.

For $m = 1$ and any $\boldsymbol{x}^{(1)} = (\boldsymbol{x}_1,) \in \triangle_M^1$, the right hand side of equation (17) can be written as

$$\sum_{j=1}^{M} (\boldsymbol{x}_1 \cdot \boldsymbol{e}_j)(\boldsymbol{e}_j,) = (\boldsymbol{x}_1,) = \boldsymbol{x}.$$

For the inductive hypothesis, suppose equation (17) holds for some arbitrary $m \geq 1$, and denote elements of $\boldsymbol{E}_M^{m+1}$ by $\boldsymbol{\eta}^{(m)} \oplus (\boldsymbol{e},)$ for some $\boldsymbol{\eta}^{(m)} \in \boldsymbol{E}_M^m$ and $\boldsymbol{e} \in \boldsymbol{E}_M$, where $\oplus$ denotes vector concatenation. Then for any $\boldsymbol{x}^{(m+1)} = \boldsymbol{x}^{(m)} \oplus (\boldsymbol{x}_{m+1},) = (\boldsymbol{x}_1, \ldots, \boldsymbol{x}_{m+1}) \in \triangle_M^{m+1}$, the right hand side of equation (17) can be written as

$$\sum_{\boldsymbol{\eta}^{(m+1)} \in \boldsymbol{E}_M^{m+1}} \left( \prod_{i=1}^{m+1} \boldsymbol{x}_i \cdot \boldsymbol{\eta}_i \right) \boldsymbol{\eta}^{(m+1)} = \sum_{\boldsymbol{\eta}^{(m)} \in \boldsymbol{E}_M^m} \sum_{j=1}^{M} \left( \prod_{i=1}^{m} \boldsymbol{x}_i \cdot \boldsymbol{\eta}_i \right) (\boldsymbol{x}_{m+1} \cdot \boldsymbol{e}_j) \boldsymbol{\eta}^{(m)} \oplus (\boldsymbol{e}_j,)$$

$$= \sum_{\boldsymbol{\eta}^{(m)} \in \boldsymbol{E}_M^m} \sum_{j=1}^{M} \left( \prod_{i=1}^{m} \boldsymbol{x}_i \cdot \boldsymbol{\eta}_i \right) (\boldsymbol{x}_{m+1} \cdot \boldsymbol{e}_j) \boldsymbol{\eta}^{(m)}$$

$$\oplus \sum_{\boldsymbol{\eta}^{(m)} \in \boldsymbol{E}_M^m} \sum_{j=1}^{M} \left( \prod_{i=1}^{m} \boldsymbol{x}_i \cdot \boldsymbol{\eta}_i \right) (\boldsymbol{x}_{m+1} \cdot \boldsymbol{e}_j)(\boldsymbol{e}_j,)$$

$$= \sum_{j=1}^{M} (\boldsymbol{x}_{m+1} \cdot \boldsymbol{e}_j) \sum_{\boldsymbol{\eta}^{(m)} \in \boldsymbol{E}_M^m} \left( \prod_{i=1}^{m} \boldsymbol{x}_i \cdot \boldsymbol{\eta}_i \right) \boldsymbol{\eta}^{(m)}$$

$$\oplus \sum_{\boldsymbol{\eta}^{(m)} \in \boldsymbol{E}_M^m} \left( \prod_{i=1}^{m} \boldsymbol{x}_i \cdot \boldsymbol{\eta}_i \right) \sum_{j=1}^{M} (\boldsymbol{x}_{m+1} \cdot \boldsymbol{e}_j)(\boldsymbol{e}_j,)$$

$$= 1 \cdot \boldsymbol{x}^{(m)} \oplus 1 \cdot (\boldsymbol{x}_{m+1},) = \boldsymbol{x}^{(m+1)},$$

where in the penultimate equality we have used the inductive hypothesis and (twice) the result of the previous induction.

We can now prove the statement of the Lemma. Applying Jensen's inequality to equation (17) with the convex function $f$, we have that

$$f(\boldsymbol{x}_1, \ldots, \boldsymbol{x}_m) = f \left( \sum_{\boldsymbol{\eta}^{(m)} \in \boldsymbol{E}_M^m} \left( \prod_{i=1}^{m} \boldsymbol{x}_i \cdot \boldsymbol{\eta}_i \right) \boldsymbol{\eta}^{(m)} \right)$$

$$\leq \sum_{\boldsymbol{\eta}^{(m)} \in \boldsymbol{E}_M^m} \left( \prod_{i=1}^{m} \boldsymbol{x}_i \cdot \boldsymbol{\eta}_i \right) f \left( \boldsymbol{\eta}^{(m)} \right).$$

Let $\boldsymbol{\mu} = \mathbb{E}[\boldsymbol{X}_1]$ denote the mean of the i.i.d. random vectors $X_i$. Then the above inequality implies

$$
\begin{aligned}
\mathbb{E}[f(\boldsymbol{X}_1, \ldots, \boldsymbol{X}_m)] &\leq \sum_{\boldsymbol{\eta}^{(m)} \in \boldsymbol{E}_M^m} \left( \prod_{i=1}^m \boldsymbol{\mu} \cdot \boldsymbol{\eta}_i \right) f\left(\boldsymbol{\eta}^{(m)}\right) \\
&= \sum_{\boldsymbol{\eta}^{(m)} \in \boldsymbol{E}_M^m} \left( \prod_{i=1}^m \mathbb{P}(\boldsymbol{X}_i' = \boldsymbol{\eta}_i) \right) f\left(\boldsymbol{\eta}^{(m)}\right) \\
&= \mathbb{E}[f(\boldsymbol{X}_1', \ldots, \boldsymbol{X}_m')].
\end{aligned}
$$

### C.3 Proof of Lemma 3

The proof of Lemma 3 itself requires two technical helping lemmas which we now state and prove.

**Lemma 4.** *For any integers $n \geq 2$ and $p \geq -1$,*

$$
\sum_{k=1}^{n-1} \frac{(n-k)^{p/2}}{\sqrt{k}} \leq n^{\frac{p+1}{2}} \int_0^1 \frac{(1-x)^{p/2}}{\sqrt{x}} dx.
$$

*Proof.* The case of $p = -1$, namely

$$
\sum_{k=1}^{n-1} \frac{1}{\sqrt{k(n-k)}} \leq \int_0^1 \frac{1}{\sqrt{x(1-x)}} dx,
$$

has already been demonstrated in Maurer [2004]. For $p > -1$, let

$$
f_p(x) := \frac{(1-x)^{p/2}}{\sqrt{x}}.
$$

We will show that each $f_p(\cdot)$ is monotonically decreasing on $(0, 1)$. Indeed,

$$
\frac{df_p}{dx}(x) = -\frac{(1-x)^{\frac{p}{2}-1}(px + 1 - x)}{2x^{3/2}} \leq -\frac{(1-x)^{p/2}}{2x^{3/2}} < 0,
$$

where for the inequalities we have used the fact that $p > -1$ and $x \in (0, 1)$. We therefore see that

$$
\begin{aligned}
\sum_{k=1}^{n-1} \frac{(n-k)^{p/2}}{\sqrt{k}} &= \sum_{k=1}^{n-1} \frac{n^{p/2}(1 - \frac{k}{n})^{p/2}}{\sqrt{n}\sqrt{\frac{k}{n}}} \\
&= n^{\frac{p+1}{2}} \sum_{k=1}^{n-1} \frac{1}{n} \frac{(1 - \frac{k}{n})^{p/2}}{\sqrt{\frac{k}{n}}} \\
&= n^{\frac{p+1}{2}} \sum_{k=1}^{n-1} \frac{1}{n} f_p\left(\frac{k}{n}\right) \\
&\leq n^{\frac{p+1}{2}} \sum_{k=1}^{n-1} \int_{\frac{k-1}{n}}^{\frac{k}{n}} f_p(x) dx \\
&= n^{\frac{p+1}{2}} \int_0^{1-\frac{1}{n}} f_p(x) dx \\
&\leq n^{\frac{p+1}{2}} \int_0^1 f_p(x) dx.
\end{aligned}
$$

$\square$

Intuitively, the proof of the above lemma works by bounding the integral below by a Riemann sum. In the following lemma we actually calculate this integral, yielding a more explicit bound on the sum in Lemma 4. We found it is easier to calculate a slightly more general integral, where the 1 in the limit and the integrand is replaced by a positive constant $a$.

**Lemma 5.** *For any real number $a > 0$ and integer $n \geq -1$,*

$$\int_0^a \frac{(a-x)^{n/2}}{\sqrt{x}}dx = \sqrt{\pi}\frac{\Gamma(\frac{n+2}{2})}{\Gamma(\frac{n+3}{2})}a^{\frac{n+1}{2}}.$$

*Proof.* Define

$$\mathrm{I}_n(a) := \int_0^a \frac{(a-x)^{n/2}}{\sqrt{x}}dx \qquad \text{and} \qquad f_n(a) := \sqrt{\pi}\frac{\Gamma(\frac{n+2}{2})}{\Gamma(\frac{n+3}{2})}a^{\frac{n+1}{2}}.$$

We proceed by induction, increasing $n$ by 2 each time. This means we need two base cases. First, for $n = -1$, we have

$$\mathrm{I}_{-1}(a) = \int_0^a \frac{1}{\sqrt{x(a-x)}}dx = \left[2\arcsin\sqrt{\frac{x}{a}}\right]_0^a = \pi = f_{-1}(a),$$

since $\Gamma(\frac{1}{2}) = \sqrt{\pi}$ and $\Gamma(1) = 1$. Second, for $n = 0$,

$$\mathrm{I}_0(a) = \int_0^a \frac{1}{\sqrt{x}}dx = \left[2\sqrt{x}\right]_0^a = 2\sqrt{a} = f_0(a),$$

since $\Gamma(\frac{3}{2}) = \frac{\sqrt{\pi}}{2}$. Now, by the Leibniz integral rule, we have

$$\frac{d}{da}\mathrm{I}_{n+2}(a) = \int_0^a \frac{\partial}{\partial a}\frac{(a-x)^{\frac{n+2}{2}}}{\sqrt{x}}dx = \frac{n+2}{2}\int_0^a \frac{(a-x)^{\frac{n}{2}}}{\sqrt{x}}dx = \frac{n+2}{2}\mathrm{I}_n(a).$$

Thus

$$\mathrm{I}_{n+2}(a) = \frac{n+2}{2}\left[\int_0^a \mathrm{I}_n(t)dt + \mathrm{I}_n(0)\right] = \frac{n+2}{2}\int_0^a \mathrm{I}_n(t)dt,$$

since $\mathrm{I}_n(0) = 0$.

Now, for the inductive step, suppose $\mathrm{I}_n(a) = f_n(a)$ for some $n \geq -1$. Then, using the previous calculation, we have

$$\begin{aligned}
\mathrm{I}_{n+2}(a) &= \frac{n+2}{2}\int_0^a f_n(t)dt \\
&= \frac{n+2}{2}\int_0^a \sqrt{\pi}\frac{\Gamma(\frac{n+2}{2})}{\Gamma(\frac{n+3}{2})}t^{\frac{n+1}{2}}dt \\
&= \sqrt{\pi}\frac{\frac{n+2}{2}\Gamma(\frac{n+2}{2})}{\frac{n+3}{2}\Gamma(\frac{n+3}{2})}a^{\frac{n+3}{2}} \\
&= \sqrt{\pi}\frac{\Gamma(\frac{n+2}{2}+1)}{\Gamma(\frac{n+3}{2}+1)}a^{\frac{n+3}{2}} \\
&= \sqrt{\pi}\frac{\Gamma\left(\frac{(n+2)+2}{2}\right)}{\Gamma\left(\frac{(n+2)+3}{2}\right)}a^{\frac{(n+2)+1}{2}} \\
&= f_{n+2}(a).
\end{aligned}$$

This completes the proof. $\square$

We are now ready to prove Lemma 3 which, for ease of reference, we restate here. For integers $M \geq 1$ and $m \geq M$,

$$\sum_{\boldsymbol{k} \in S_{m,M}^{>0}} \frac{1}{\prod_{j=1}^M \sqrt{k_j}} \leq \frac{\pi^{\frac{M}{2}}m^{\frac{M-2}{2}}}{\Gamma(\frac{M}{2})}.$$

*Proof.* (of Lemma 3) We proceed by induction on $M$. For $M = 1$, the set $S_{m,M}$ contains a single element, namely the one-dimensional vector $\boldsymbol{k} = (k_1,) = (m,)$. In this case, the left hand side is $1/\sqrt{m}$ while the right hand side is $\sqrt{\pi}/(\sqrt{m}\Gamma(1/2)) = 1/\sqrt{m}$, since $\Gamma(1/2) = \sqrt{\pi}$.

Now, as the inductive hypothesis, assume the inequality of Lemma 3 holds for some fixed $M \geq 1$ and all $m \geq M$. Then for all $m \geq M + 1$, we have

$$
\sum_{\boldsymbol{k} \in S_{m,M+1}^{>0}} \frac{1}{\prod_{j=1}^{M+1} \sqrt{k_j}} = \sum_{k_1=1}^{m-M} \frac{1}{\sqrt{k_1}} \sum_{\boldsymbol{k'} \in S_{m-k_1,M}^{>0}} \frac{1}{\prod_{j=1}^{M} \sqrt{k'_j}}
$$

$$
\leq \sum_{k_1=1}^{m-M} \frac{1}{\sqrt{k_1}} \frac{\pi^{\frac{M}{2}}(m-k_1)^{\frac{M-2}{2}}}{\Gamma(\frac{M}{2})} \qquad \text{(by the inductive hypothesis)}
$$

$$
= \frac{\pi^{\frac{M}{2}}}{\Gamma(\frac{M}{2})} \sum_{k_1=1}^{m-M} \frac{(m-k_1)^{\frac{M-2}{2}}}{\sqrt{k_1}}
$$

$$
\leq \frac{\pi^{\frac{M}{2}}}{\Gamma(\frac{M}{2})} \sum_{k_1=1}^{m-1} \frac{(m-k_1)^{\frac{M-2}{2}}}{\sqrt{k_1}} \qquad \text{(enlarging the sum domain)}
$$

$$
\leq \frac{\pi^{\frac{M}{2}}}{\Gamma(\frac{M}{2})} m^{\frac{M-1}{2}} \int_0^1 \frac{(1-x)^{\frac{M-2}{2}}}{\sqrt{x}} dx \qquad \text{(by Lemma 4)}
$$

$$
= \frac{\pi^{\frac{M}{2}}}{\Gamma(\frac{M}{2})} m^{\frac{M-1}{2}} \sqrt{\pi} \frac{\Gamma(\frac{M}{2})}{\Gamma(\frac{M+1}{2})} \qquad \text{(by Lemma 5)}
$$

$$
= \frac{\pi^{\frac{M+1}{2}} m^{\frac{M-1}{2}}}{\Gamma(\frac{M+1}{2})},
$$

as required. $\qquad\square$

### C.4   Proof of Proposition 1

*Proof.* The case where $q_j = 1$ or $p_j = 1$ can be dealt with trivially by splitting into the three following subcases

- $q_j = p_j = 1 \implies \text{kl}(q_j\|p_j) = \text{kl}(\boldsymbol{q}\|\boldsymbol{p}) = 0$
- $q_j = 1, p_j \neq 1 \implies \text{kl}(q_j\|p_j) = \text{kl}(\boldsymbol{q}\|\boldsymbol{p}) = -\log p_j$
- $q_j \neq 1, p_j = 1 \implies \text{kl}(q_j\|p_j) = \text{kl}(\boldsymbol{q}\|\boldsymbol{p}) = \infty.$

For $q_j \neq 1$ and $p_j \neq 1$ define the distributions $\tilde{\boldsymbol{q}}, \tilde{\boldsymbol{p}} \in \triangle_M$ by $\tilde{q}_j = \tilde{p}_j = 0$ and

$$
\tilde{q}_i = \frac{q_i}{1-q_j} \qquad \text{and} \qquad \tilde{p}_i = \frac{p_i}{1-p_j}
$$

for $i \neq j$. Then

$$
\sum_{i \neq j} q_i \log \frac{q_i}{p_i} = \sum_{i \neq j} (1-q_j)\tilde{q}_i \log \frac{(1-q_j)\tilde{q}_i}{(1-p_j)\tilde{p}_i}
$$

$$
= (1-q_j) \sum_{i \neq j} \tilde{q}_i \log \frac{\tilde{q}_i}{\tilde{p}_i} + \tilde{q}_i \log \frac{1-q_j}{1-p_j}
$$

$$
= (1-q_j)\text{kl}(\tilde{\boldsymbol{q}}\|\tilde{\boldsymbol{p}}) + (1-q_j) \log \frac{1-q_j}{1-p_j}
$$

$$
\geq (1-q_j) \log \frac{1-q_j}{1-p_j}.
$$

The final inequality holds since $\mathrm{kl}(\tilde{q}\|\tilde{p}) \geq 0$. Further, note that we have equality if and only if $\tilde{q} = \tilde{p}$, which, by their definitions, translates to

$$p_i = \frac{1 - p_j}{1 - q_j} q_i$$

for all $i \neq j$. If we now add $q_j \log \frac{q_j}{p_j}$ to both sides, we obtain

$$\mathrm{kl}(\boldsymbol{q}\|\boldsymbol{p}) \geq (1 - q_j) \log \frac{1 - q_j}{1 - p_j} + q_j \log \frac{q_j}{p_j} = \mathrm{kl}(q_j\|p_j),$$

with the same condition for equality. $\qquad\square$

The following proposition makes more precise the argument found at the beginning of Section 4 for how Proposition 1 can be used to derive the tightest possible lower and upper bounds on each $R_D^j(Q)$.

**Proposition 5.** *Suppose that $\boldsymbol{q}, \boldsymbol{p} \in \triangle_M$ are such that $\mathrm{kl}(\boldsymbol{q}\|\boldsymbol{p}) \leq B$, where $\boldsymbol{q}$ is known and $\boldsymbol{p}$ is unknown. Then, in the absence of any further information, the tightest bound that can be obtained on each $p_j$ is*

$$p_j \leq \mathrm{kl}^{-1}(q_j, B).$$

*Proof.* Suppose $p_j > \mathrm{kl}^{-1}(q_j, B)$. Then, by definition of $\mathrm{kl}^{-1}$, we have that $\mathrm{kl}(q_j\|p_j) > B$. By Proposition 1, this would then imply $\mathrm{kl}(\boldsymbol{q}\|\boldsymbol{p}) > B$, contradicting our assumption. Therefore $p_j \leq \mathrm{kl}^{-1}(q_j, B)$. Now, with the information we have, we cannot rule out that

$$p_i = \frac{1 - p_j}{1 - q_j} q_i$$

for all $i \neq j$ and thus, by Proposition 1, that $\mathrm{kl}(q_j\|p_j) = \mathrm{kl}(\boldsymbol{q}\|\boldsymbol{p})$. Further, we cannot rule out that $\mathrm{kl}(\boldsymbol{q}\|\boldsymbol{p}) = B$. Thus, it is possible that $\mathrm{kl}(q_j\|p_j) = B$, in which case $p_j = \mathrm{kl}^{-1}(q_j, B)$. We therefore see that $\mathrm{kl}^{-1}(q_j, B)$ is the tightest possible upper bound on $p_j$, for each $j \in [M]$. $\qquad\square$

### C.5 Proof of Theorem 2

Before proving the proposition, we first argue that $\mathrm{kl}_{\boldsymbol{\ell}}^{-1}(\boldsymbol{u}|c)$ given by Definition 1 is well-defined. First, note that $A_{\boldsymbol{u}} := \{\boldsymbol{v} \in \triangle_M : \mathrm{kl}(\boldsymbol{u}\|\boldsymbol{v}) \leq c\}$ is compact (boundedness is clear and it is closed because it is the preimage of the closed set $[0, c]$ under the continuous map $\boldsymbol{v} \mapsto \mathrm{kl}(\boldsymbol{u}\|\boldsymbol{v})$) and so the continuous function $f_{\boldsymbol{\ell}}$ achieves its supremum on $A_{\boldsymbol{u}}$. Further, note that $A_{\boldsymbol{u}}$ is a convex subset of $\triangle_M$ (because the map $\boldsymbol{v} \mapsto \mathrm{kl}(\boldsymbol{u}\|\boldsymbol{v})$ is convex) and $f_{\boldsymbol{\ell}}$ is linear, so the supremum of $f_{\boldsymbol{\ell}}$ over $A_{\boldsymbol{u}}$ is achieved and is located on the boundary of $A_{\boldsymbol{u}}$. This means we can replace the inequality constraint $\mathrm{kl}(\boldsymbol{u}\|\boldsymbol{v}) \leq c$ in Definition 1 with the equality constraint $\mathrm{kl}(\boldsymbol{u}\|\boldsymbol{v}) = c$. Finally, if $\boldsymbol{u} \in \triangle_M^{>0}$ then $A_{\boldsymbol{u}}$ is a *strictly* convex subset of $\triangle_M$ (because the map $\boldsymbol{v} \mapsto \mathrm{kl}(\boldsymbol{u}\|\boldsymbol{v})$ is then *strictly* convex) and so the supremum of $f_{\boldsymbol{\ell}}$ occurs at a *unique* point on the boundary of $A_{\boldsymbol{u}}$. In other words, if $\boldsymbol{u} \in \triangle_M^{>0}$ then $\mathrm{kl}_{\boldsymbol{\ell}}^{-1}(\boldsymbol{u}|c)$ is defined *uniquely*.

We now prove Theorem 2. While our proof technique is somewhat analogous to the technique used in Clerico et al. [2022] to obtain derivatives of the one-dimensional kl-inverse, our theorem directly yields derivatives on the total risk by (implicitly) employing the envelope theorem (see for example Takayama and Akira, 1985).

**Proof Outline:** We first derive the expression given for $\boldsymbol{v}^*(\tilde{\boldsymbol{u}}) = \mathrm{kl}_{\boldsymbol{\ell}}^{-1}(\boldsymbol{u}|c)$ given on line (5) of the theorem using the method of Lagrange multipliers. Since we are working on the simplex, we make things easier for ourselves by first making the substitution $t_j = \ln v_j$ to make the $v_j > 0$ constraints unnecessary. The method of Lagrange multipliers yields both the maximum and the minimum (recall that $\mathrm{kl}_{\boldsymbol{\ell}}^{-1}(\boldsymbol{u}|c)$ is defined as the location of a maximum) for the two values of the Lagrange multiplier $\mu$. We show that exactly one of these values lies in the interval $\mu \in (-\infty, -\max_j \ell_j)$ and that this one corresponds to the maximum. This shows that the value $\mu^*$ Theorem 2 instructs us to find indeed yields $\boldsymbol{v}^*(\tilde{\boldsymbol{u}}) = \mathrm{kl}_{\boldsymbol{\ell}}^{-1}(\boldsymbol{u}|c)$. Finally, we derive the partial derivatives of $\mathrm{kl}_{\boldsymbol{\ell}}^{-1}(\boldsymbol{u}|c)$ with respect the $\tilde{u}_j$ to obtain the second part of the theorem, namely line (6) by employing the envelope theorem.

*Proof.* (of Theorem 2) We start by deriving the implicit expression for $v^*(\tilde{u}) = \mathrm{kl}_{\ell}^{-1}(u|c)$ given in the proposition by solving a transformed version of the optimisation problem given by Definition 1 using the method of Lagrange multipliers. We obtain two solutions to the Lagrangian equations, which must correspond to the maximum and minimum total risk over the set $A_u := \{v \in \triangle_M : \mathrm{kl}(u\|v) \leq c\}$ because, as argued in the main text (see the discussion after Definition 1), $A_u$ is compact and so the linear total risk $f_{\ell}(v)$ attains its maximum and minimum on $A_u$.

By definition of $v^*(\tilde{u}) = \mathrm{kl}_{\ell}^{-1}(u|c)$, we know that $\mathrm{kl}(v^*(\tilde{u})\|u) \leq c$. Since, by assumption, $u_j > 0$ for all $j$, we see that $v^*(\tilde{u})_j > 0$ for all $j$, otherwise we would have $\mathrm{kl}(v^*(\tilde{u})\|u) = \infty$, a contradiction. Thus $v^*(\tilde{u}) \in \triangle_M^{>0}$ and we are permitted to instead optimise over the unbounded variable $t \in \mathbb{R}^M$, where $t_j := \ln v_j$. With this transformation, the constraint $v \in \triangle_M$ can be replaced simply by $\sum_j e^{t_j} = 1$ and the optimisation problem becomes

$$\text{Maximise:} \quad F(t) := \sum_{j=1}^{M} \ell_j e^{t_j}$$

$$\text{Subject to:} \quad g(t; u, c) := \mathrm{kl}(u\|e^t) - c = 0,$$

$$h(t) := \sum_{j=1}^{M} e^{t_j} - 1 = 0,$$

where $e^t \in \mathbb{R}^M$ is defined by $(e^t)_j := e^{t_j}$. Note that $F(t) = f_{\ell}(e^t)$. Following the terminology of mathematical economics, we call the $t_j$ the *optimisation variables*, and the $\tilde{u}_j$ (namely the $u_j$ and $c$) the *choice variables*. The vector $\ell$ is considered fixed—we neither want to optimise over it nor differentiate with respect to it—which is why we occasionally suppress it from the notation henceforth.

For each $\tilde{u}$, let $v^*(\tilde{u})$ and $t^*(\tilde{u})$ be the solutions to the original and transformed optimisation problems respectively. Since the map $v = e^t$ is one-to-one, it is clear that since $v^*(\tilde{u})$ exists uniquely, so does $t^*(\tilde{u})$, and that they are related by $v^*(\tilde{u}) = e^{t^*(\tilde{u})}$. We therefore have the identity

$$f_{\ell}(v^*(\tilde{u})) \equiv F(t^*(\tilde{u})).$$

Recalling that $f_{\ell}^*(\tilde{u}) := f_{\ell}(v^*(\tilde{u}))$, we see that

$$\nabla_{\tilde{u}} f_{\ell}^*(\tilde{u}) \equiv \nabla_{\tilde{u}} F(t^*(\tilde{u})). \tag{18}$$

the derivatives of $f_{\ell}(\mathrm{kl}_{\ell}^{-1}(u|c))$ with respect to $u$ and $c$ are given by $\nabla_{\tilde{u}} F(t^*(\tilde{u}))$.

Using the method of Lagrange multipliers, there exist real numbers $\lambda^* = \lambda^*(\tilde{u})$ and $\mu^* = \mu^*(\tilde{u})$ such that $(t^*, \lambda^*, \mu^*)$ is a stationary point (with respect to $t$, $\lambda$ and $\mu$) of the Lagrangian function

$$\mathcal{L}(t, \lambda, \mu; \tilde{u}) := F(t) + \lambda g(t; \tilde{u}) + \mu h(t).$$

Let $F_t(\cdot)$ and $h_t(\cdot)$ denote the gradient vectors of $F$ and $h$ respectively, and let $g_t(\cdot\,; \tilde{u})$ and $g_{\tilde{u}}(t; \cdot)$ denote the gradient vectors of $g$ with respect to $t$ only and $\tilde{u}$ only, respectively. Simple calculation yields

$$g_t(t; \tilde{u}) = \left( \frac{\partial g}{\partial t_1}(t; \tilde{u}), \dots, \frac{\partial g}{\partial t_M}(t; \tilde{u}) \right) = -u \quad \text{and}$$

$$g_{\tilde{u}}(t; \tilde{u}) = \left( \frac{\partial g}{\partial \tilde{u}_1}(t; \tilde{u}), \dots, \frac{\partial g}{\partial \tilde{u}_{M+1}}(t; \tilde{u}) \right) = \left( 1 - t_1 + \log u_1, \dots, 1 - t_M + \log u_M, -1 \right). \tag{19}$$

Then, taking the partial derivatives of $\mathcal{L}$ with respect to $\lambda$, $\mu$ and the $t_j$, we have that $(t, \lambda, \mu) = (t^*(\tilde{u}), \lambda^*(\tilde{u}), \mu^*(\tilde{u}))$ solves the simultaneous equations

$$F_t(t) + \lambda g_t(t; \tilde{u}) + \mu h_t(t) = 0, \tag{20}$$

$$g(t; \tilde{u}) = 0, \quad \text{and}$$

$$h(t) = 0,$$

where the last two equations recover the constraints. Substituting the gradients $F_t$, $g_t$ and $h_t$, the first equation reduces to

$$\ell \odot e^t - \lambda u + \mu e^t = 0,$$

which implies that for all $j \in [M]$

$$e^{t_j} = \frac{\lambda u_j}{\mu + \ell_j}.$$
(21)

Substituting this into the constraints $g = h = 0$ yields the following simultaneous equations in $\lambda$ and $\mu$

$$c = \mathrm{kl}(\boldsymbol{u} \| e^{\boldsymbol{t}}) = \sum_{j=1}^{M} u_j \log \frac{u_j}{e^{t_j}} = \sum_{j=1}^{M} u_j \log \frac{\mu + \ell_j}{\lambda} \quad \text{and} \quad \lambda \sum_{j=1}^{M} \frac{u_j}{\mu + \ell_j} = 1.$$

Substituting the second into the first and rearranging the second, this is equivalent to solving

$$c = \sum_{j=1}^{M} u_j \log \left( (\mu + \ell_j) \sum_{k=1}^{M} \frac{u_k}{\mu + \ell_k} \right) \quad \text{and} \quad \lambda = \left( \sum_{j=1}^{M} \frac{u_j}{\mu + \ell_j} \right)^{-1}.$$
(22)

It has already been established in the discussion after Definition 1 that $f_{\boldsymbol{\ell}}(\boldsymbol{v})$ attains its maximum on the set $A_{\boldsymbol{u}} := \{\boldsymbol{v} \in \triangle_M : \mathrm{kl}(\boldsymbol{u} \| \boldsymbol{v}) \le c\}$. Therefore $F(\boldsymbol{t})$ also attains its maximum on $\mathbb{R}^M$ and one of the solutions to these simultaneous equations corresponds to this maximum. We first show that there is a single solution to the first equation in the set $(-\infty, -\max_j \ell_j)$, referred to as $\mu^*(\tilde{\boldsymbol{u}})$ in the proposition. Second, we show that any other solution corresponds to a smaller total risk, so that $\mu^*(\tilde{\boldsymbol{u}})$ corresponds to the maximum total risk and yields $\boldsymbol{v}^*(\tilde{\boldsymbol{u}}) = \mathrm{kl}_{\boldsymbol{\ell}}^{-1}(\boldsymbol{u}|c)$ when $\mu^*(\tilde{\boldsymbol{u}})$ and the associated $\lambda^*(\tilde{\boldsymbol{u}})$ are substituted into Equation 21.

For the first step, note that since the $e^{t_j}$ are probabilities, we see from Equation 21 that either $\mu + \ell_j > 0$ for all $j$ (in the case that $\lambda > 0$), or $\mu + \ell_j < 0$ for all $j$ (in the case that $\lambda < 0$). Thus any solutions $\mu$ to the first equation must be in $(-\infty, -\max_j \ell_j)$ or $(-\min_j \ell_j, \infty)$. If $\mu \in (-\infty, -\max_j \ell_j)$ then the first equation can be written as $c = \phi_{\boldsymbol{\ell}}(\mu)$, with $\phi_{\boldsymbol{\ell}}$ as defined in the statement of the proposition. We now show that $\phi_{\boldsymbol{\ell}}$ is strictly increasing in $\mu$, and that $\phi_{\boldsymbol{\ell}}(\mu) \to 0$ as $\mu \to -\infty$ and $\phi_{\boldsymbol{\ell}}(\mu) \to \infty$ as $\mu \to -\max_j \ell_j$, so that $c = \phi_{\boldsymbol{\ell}}(\mu)$ does indeed have a single solution in the set $(-\infty, -\max_j \ell_j)$. Straightforward differentiation and algebra shows that

$$\phi_{\boldsymbol{\ell}}'(\mu) = \sum_{j=1}^{M} \frac{u_j}{(\mu + \ell_j) \sum_{k=1}^{M} \frac{u_k}{\mu + \ell_k}} \left( \sum_{k'=1}^{M} \frac{u_{k'}}{\mu + \ell_{k'}} - (\mu + \ell_j) \sum_{k'=1}^{M} \frac{u_{k'}}{(\mu + \ell_{k'})^2} \right)$$

$$= \frac{\left( \sum_{j=1}^{M} \frac{u_j}{\mu + \ell_j} \right)^2 - \sum_{j=1}^{M} \frac{u_j}{(\mu + \ell_j)^2}}{\sum_{k=1}^{M} \frac{u_k}{\mu + \ell_k}}.$$

Jensen's inequality demonstrates that the numerator is strictly negative, where strictness is due to the assumption that the $\ell_j$ are not all equal. Further, since the denominator is strictly negative (since we are dealing with the case where $\mu \in (-\infty, -\max_j \ell_j)$), we see that $\phi_{\boldsymbol{\ell}}$ is strictly increasing for $\mu \in (-\infty, -\max_j \ell_j)$.[5] Turning to the limits, we first show that $\phi_{\boldsymbol{\ell}}(\mu) \to \infty$ as $\mu \to -\max_j \ell_j$.

We now determine the left hand limit. Define $J = \{j \in [M] : \ell_j = \max_k \ell_k\}$, noting that this is a strict subset of $[M]$ since by assumption the $\ell_j$ are not all equal. We then have that for

---

[5]Incidentally, this argument also shows that there is at most one solution to the first equation in (22) in the range $(-\min_j \ell_j, \infty)$. There indeed exists a unique solution, which corresponds to the minimum total risk, but we do not prove this.

$\mu \in (-\infty, \max_j \ell_j)$

$$e^{\phi_{\ell}(\mu)} = \left( -\sum_{j=1}^{M} \frac{u_j}{\mu + \ell_j} \right) \left( \prod_{k=1}^{M} \left( -(\mu + \ell_k) \right)^{u_k} \right)$$

$$= \left( -\sum_{j \in J} \frac{u_j}{\mu + \ell_j} - \sum_{j' \notin J} \frac{u_{j'}}{\mu + \ell_{j'}} \right) \prod_{k \in J} \left( -(\mu + \ell_k) \right)^{u_k} \prod_{k' \notin J} \left( -(\mu + \ell_{k'}) \right)^{u_{k'}}$$

$$\geq \left( -\sum_{j \in J} \frac{u_j}{\mu + \ell_j} \right) \prod_{k \in J} \left( -(\mu + \ell_k) \right)^{u_k} \prod_{k' \notin J} \left( -(\mu + \ell_{k'}) \right)^{u_{k'}}$$

$$= \frac{\left( \sum_{j \in J} u_j \right) \left( \prod_{k' \notin J} \left( -(\mu + \ell_{k'}) \right)^{u_{k'}} \right)}{\left( -(\mu + \max_j \ell_j) \right)^{1 - \sum_{k \in J} u_k}}.$$

The first term in the numerator is a positive constant, independent of $\mu$. The second term in the numerator tends to a finite positive limit as $\mu \uparrow -\max_j \ell_j$. Since $[M] \setminus J$ is non-empty, the power in the denominator is positive and the term in the outer brackets is positive and tends to zero as $\mu \uparrow -\max_j \ell_j$. Thus $e^{\phi_{\ell}(\mu)} \to \infty$ as $\mu \uparrow -\max_j \ell_j$ and, by the continuity of the logarithm, $\phi_{\ell}(\mu)$ as $\mu \uparrow -\max_j \ell_j$.

We now determine $\lim_{\mu \to -\infty} \phi_{\ell}(\mu)$ by sandwiching $\phi(\mu)$ between two functions that both tend to zero as $\mu \to -\infty$. First, since $\ell_j \geq 0$ for all $j$, for $\mu \in (-\infty, -\max_j \ell_j)$ we have

$$\log \left( -\sum_{j=1}^{M} \frac{u_j}{\mu + \ell_j} \right) \geq \log \left( -\sum_{j=1}^{M} \frac{u_j}{\mu} \right) = -\log(-\mu) = -\sum_{j=1}^{M} u_j \log(-\mu),$$

and so

$$\phi_{\ell}(\mu) \geq -\sum_{j=1}^{M} u_j \log(-\mu) + \sum_{j=1}^{M} u_j \log \left( -(\mu + \ell_j) \right) = \sum_{j=1}^{M} u_j \log \left( 1 + \frac{\ell_j}{\mu} \right) \to 0 \quad \text{as} \quad \mu \to -\infty.$$

Similarly,

$$\sum_{j=1}^{M} u_j \log \left( -(\mu + \ell_j) \right) \leq \sum_{j=1}^{M} u_j \log(-\mu) = \log(-\mu),$$

and so

$$\phi_{\ell}(\mu) \leq \log \left( \mu \sum_{j=1}^{M} \frac{u_j}{\mu + \ell_j} \right) = \log \left( \sum_{j=1}^{M} \frac{u_j}{1 + \frac{\ell_j}{\mu}} \right) \to 0 \quad \text{as} \quad \mu \to -\infty.$$

This completes the first step, namely showing that there does indeed exist a unique solution $\mu^*(\tilde{\boldsymbol{u}})$ in the set $(-\ell_1, \infty)$ to the first equation in line (22).

We now turn to the second step, namely showing that this solution corresponds to the maximum total risk. Given a value of the Lagrange multiplier $\mu$, substitution into Equation 21 gives

$$e^{t_j}(\mu) = \frac{\frac{u_j}{\mu + \ell_j}}{\sum_{k=1}^{M} \frac{u_k}{\mu + \ell_k}}$$

and therefore total risk

$$R(\mu) = \frac{\sum_{j=1}^{M} \frac{u_j \ell_j}{\mu + \ell_j}}{\sum_{k=1}^{M} \frac{u_k}{\mu + \ell_k}}.$$

To prove that the solution $\mu^*(\tilde{\boldsymbol{u}}) \in (-\infty, -\max_j \ell_j)$ is the solution to the first equation in line (22) that maximises $R$, it suffices to show that $R(\mu) \to \sum_{j=1}^{M} u_j \ell_j$ as $|\mu| \to \infty$ and $R'(\mu) \geq 0$ for all $\mu \in (-\infty, -\max_j \ell_j) \cup (-\min_j \ell_j, \infty)$, so that

$$\inf_{\mu \in (-\infty, -\max_j \ell_j)} R(\mu) \geq \sup_{\mu \in (-\min_j \ell_j, \infty)} R(\mu).$$

This suffices as we have already proved that $\mu^*(\tilde{\boldsymbol{u}})$ is the only solution in $(-\infty, -\max_j \ell_j)$ to the first equation in line (22), and that no solutions exists in the set $[-\max_j \ell_j, -\min_j \ell_j]$.

The limit can be easily evaluated by first rewriting $R(\mu)$ and then taking the limit as $|\mu| \to \infty$ as follows

$$R(\mu) = \frac{\sum_{j=1}^{M} \frac{u_j \ell_j}{1 + \frac{\ell_j}{\mu}}}{\sum_{k=1}^{M} \frac{u_k}{1 + \frac{\ell_k}{\mu}}} \to \frac{\sum_{j=1}^{M} u_j \ell_j}{\sum_{k=1}^{M} u_k} = \sum_{j=1}^{M} u_j \ell_j.$$

To show that $R'(\mu) \geq 0$, let $\ell_{(j)}$ denote the $j$'th smallest component of $\boldsymbol{\ell}$ (breaking ties arbitrarily), so that $\ell_{(1)} \leq \cdots \leq \ell_{(M)}$, and use the quotient rule to see that

$$R'(\mu) \geq 0 \iff \frac{\left(\sum_{k=1}^{M} \frac{u_k}{\mu + \ell_k}\right)\left(\sum_{j=1}^{M} \frac{-u_j \ell_j}{(\mu + \ell_j)^2}\right) - \left(\sum_{j=1}^{M} \frac{u_j \ell_j}{\mu + \ell_j}\right)\left(\sum_{k=1}^{M} \frac{-u_k}{(\mu + \ell_k)^2}\right)}{\left(\sum_{p=1}^{M} \frac{u_p}{\mu + \ell_p}\right)^2} \geq 0$$

$$\iff \sum_{j=1}^{M}\sum_{k=1}^{M} \frac{u_j u_k \ell_j}{(\mu + \ell_j)(\mu + \ell_k)}\left(\frac{1}{\mu + \ell_k} - \frac{1}{\mu + \ell_j}\right) \geq 0$$

$$\iff \sum_{\substack{j,k \in [M] \\ k < j}} \frac{u_j u_k \ell_{(j)}}{(\mu + \ell_{(j)})(\mu + \ell_{(k)})}\left(\frac{1}{\mu + \ell_{(k)}} - \frac{1}{\mu + \ell_{(j)}}\right)$$

$$+ \sum_{\substack{j,k \in [M] \\ k > j}} \frac{u_j u_k \ell_{(j)}}{(\mu + \ell_{(j)})(\mu + \ell_{(k)})}\left(\frac{1}{\mu + \ell_{(k)}} - \frac{1}{\mu + \ell_{(j)}}\right) \geq 0,$$

where in the final line we have dropped the summands where $k = j$ since they equal zero as the terms in the bracket cancel. This final inequality holds since the first sum can be bounded below by the negative of the second sum as follows

$$\sum_{\substack{j,k \in [M] \\ k < j}} \frac{u_j u_k \ell_{(j)}}{(\mu + \ell_{(j)})(\mu + \ell_{(k)})}\left(\frac{1}{\mu + \ell_{(k)}} - \frac{1}{\mu + \ell_{(j)}}\right)$$

$$\geq \sum_{\substack{j,k \in [M] \\ k < j}} \frac{u_j u_k \ell_{(k)}}{(\mu + \ell_{(j)})(\mu + \ell_{(k)})}\left(\frac{1}{\mu + \ell_{(k)}} - \frac{1}{\mu + \ell_{(j)}}\right) \quad \text{(since } \ell_{(k)} \leq \ell_{(j)} \text{ for } k < j\text{)}$$

$$= \sum_{\substack{j,k \in [M] \\ k > j}} \frac{u_k u_j \ell_{(j)}}{(\mu + \ell_{(k)})(\mu + \ell_{(j)})}\left(\frac{1}{\mu + \ell_{(j)}} - \frac{1}{\mu + \ell_{(k)}}\right) \quad \text{(swapping dummy variables } j, k\text{)}.$$

We now turn to finding the partial derivatives of $F(\boldsymbol{t}^*(\tilde{\boldsymbol{u}}))$ with respect the $\tilde{u}_j$, which in turn will allow us to find the partial derivatives of $\text{kl}_{\boldsymbol{\ell}}^{-1}(\boldsymbol{u}|c)$. Let $\nabla_{\tilde{\boldsymbol{u}}}$ denote the gradient operator with respect to $\tilde{\boldsymbol{u}}$. Then the quantity we are after is $\nabla_{\tilde{\boldsymbol{u}}} F(\boldsymbol{t}^*(\tilde{\boldsymbol{u}})) \in \mathbb{R}^{M+1}$, the $j$'th component of which is

$$\left(\nabla_{\tilde{\boldsymbol{u}}} F(\boldsymbol{t}^*(\tilde{\boldsymbol{u}}))\right)_j = \sum_{k=1}^{M+1} \frac{\partial F}{\partial t_k}(\boldsymbol{t}^*(\tilde{\boldsymbol{u}})) \frac{\partial t_k^*}{\partial \tilde{u}_j}(\tilde{\boldsymbol{u}}) = F_{\boldsymbol{t}}(\boldsymbol{t}^*(\tilde{\boldsymbol{u}})) \cdot \frac{\partial \boldsymbol{t}^*}{\partial \tilde{u}_j}(\tilde{\boldsymbol{u}}) \in \mathbb{R}.$$

Thus the full gradient vector is

$$\nabla_{\tilde{\boldsymbol{u}}} F(\boldsymbol{t}^*(\tilde{\boldsymbol{u}})) = F_{\boldsymbol{t}}(\boldsymbol{t}^*(\tilde{\boldsymbol{u}})) \nabla_{\tilde{\boldsymbol{u}}} \boldsymbol{t}^*(\tilde{\boldsymbol{u}}), \tag{23}$$

where $\nabla_{\tilde{\boldsymbol{u}}} \boldsymbol{t}^*(\tilde{\boldsymbol{u}})$ is the $M \times (M + 1)$ matrix given by

$$\left(\nabla_{\tilde{\boldsymbol{u}}} \boldsymbol{t}^*(\tilde{\boldsymbol{u}})\right)_{j,k} = \frac{\partial t_k^*}{\partial \tilde{u}_j}(\tilde{\boldsymbol{u}}).$$

Finding an expression for this matrix is difficult. Fortunately we can avoid needing to by using a trick from mathematical economics referred to as the envelope theorem, as we now show.

First, note that since, for all $\tilde{\boldsymbol{u}}$, the constraints $g = h = 0$ are satisfied by $\boldsymbol{t}^*(\tilde{\boldsymbol{u}})$, we have the identities

$$g(\boldsymbol{t}^*(\tilde{\boldsymbol{u}}), \tilde{\boldsymbol{u}}) \equiv 0 \quad \text{and} \quad h(\boldsymbol{t}^*(\tilde{\boldsymbol{u}})) \equiv 0.$$

Differentiating these identities with respect to $\tilde{u}_j$ then yields

$$g_{\boldsymbol{t}}(\boldsymbol{t}^*(\tilde{\boldsymbol{u}}), \tilde{\boldsymbol{u}}) \cdot \frac{\partial \boldsymbol{t}^*}{\partial \tilde{u}_j}(\tilde{\boldsymbol{u}}) + g_{\tilde{u}_j}(\boldsymbol{t}^*(\tilde{\boldsymbol{u}}), \tilde{\boldsymbol{u}}) \equiv 0 \quad \text{and} \quad h_{\boldsymbol{t}}(\boldsymbol{t}^*(\tilde{\boldsymbol{u}})) \cdot \frac{\partial \boldsymbol{t}^*}{\partial \tilde{u}_j}(\tilde{\boldsymbol{u}}) \equiv 0.$$

As before, we can write these $M + 1$ pairs of equations as the following pair of matrix equations

$$g_{\boldsymbol{t}}(\boldsymbol{t}^*(\tilde{\boldsymbol{u}}), \tilde{\boldsymbol{u}}) \nabla_{\tilde{\boldsymbol{u}}} \boldsymbol{t}^*(\tilde{\boldsymbol{u}}) + g_{\tilde{\boldsymbol{u}}}(\boldsymbol{t}^*(\tilde{\boldsymbol{u}}), \tilde{\boldsymbol{u}}) \equiv \boldsymbol{0} \quad \text{and} \quad h_{\boldsymbol{t}}(\boldsymbol{t}^*(\tilde{\boldsymbol{u}})) \nabla_{\tilde{\boldsymbol{u}}} \boldsymbol{t}^*(\tilde{\boldsymbol{u}}) \equiv \boldsymbol{0}.$$

Multiplying these identities by $\lambda^*(\tilde{\boldsymbol{u}})$ and $\mu^*(\tilde{\boldsymbol{u}})$ respectively, and combining with equation (23), yields

$$\nabla_{\tilde{\boldsymbol{u}}} F(\boldsymbol{t}^*(\tilde{\boldsymbol{u}})) = \Big( F_{\boldsymbol{t}}(\boldsymbol{t}^*(\tilde{\boldsymbol{u}})) + \lambda^*(\tilde{\boldsymbol{u}}) g_{\boldsymbol{t}}(\boldsymbol{t}^*(\tilde{\boldsymbol{u}}), \tilde{\boldsymbol{u}}) + \mu^*(\tilde{\boldsymbol{u}}) h_{\boldsymbol{t}}(\boldsymbol{t}^*(\tilde{\boldsymbol{u}})) \Big) \nabla_{\tilde{\boldsymbol{u}}} \boldsymbol{t}^*(\tilde{\boldsymbol{u}})$$
$$+ \lambda^*(\tilde{\boldsymbol{u}}) g_{\tilde{\boldsymbol{u}}}(\boldsymbol{t}^*(\tilde{\boldsymbol{u}}), \tilde{\boldsymbol{u}})$$
$$= \lambda^*(\tilde{\boldsymbol{u}}) g_{\tilde{\boldsymbol{u}}}(\boldsymbol{t}^*(\tilde{\boldsymbol{u}}), \tilde{\boldsymbol{u}}),$$

where the final equality comes from noting that the terms in the large bracket vanish due to equation (20). Recalling the expression for $g_{\tilde{\boldsymbol{u}}}(\boldsymbol{t}; \tilde{\boldsymbol{u}})$ given by Equation 19 and that $\boldsymbol{v}^*(\tilde{\boldsymbol{u}}) = \exp(\boldsymbol{t}^*(\tilde{\boldsymbol{u}}))$ we obtain

$$\nabla_{\tilde{\boldsymbol{u}}} F(\boldsymbol{t}^*(\tilde{\boldsymbol{u}})) = \lambda^*(\tilde{\boldsymbol{u}}) \Big( 1 - \boldsymbol{t}^*(\tilde{\boldsymbol{u}})_1 + \log u_1, \ldots, 1 - \boldsymbol{t}^*(\tilde{\boldsymbol{u}})_M + \log u_M, -1 \Big)$$
$$= \lambda^*(\tilde{\boldsymbol{u}}) \left( 1 + \log \frac{u_1}{\boldsymbol{v}^*(\tilde{\boldsymbol{u}})_1}, \ldots, 1 + \log \frac{u_M}{\boldsymbol{v}^*(\tilde{\boldsymbol{u}})_M}, -1 \right)$$

Finally, recalling Equivalence (18), namely $\nabla_{\tilde{\boldsymbol{u}}} f_\ell^*(\tilde{\boldsymbol{u}}) \equiv \nabla_{\tilde{\boldsymbol{u}}} F(\boldsymbol{t}^*(\tilde{\boldsymbol{u}}))$, we see that the above expression gives the derivatives $\frac{\partial f_\ell^*}{\partial u_j}(\tilde{\boldsymbol{u}})$ and $\frac{\partial f_\ell^*}{\partial c}(\tilde{\boldsymbol{u}})$ stated in the proposition, thus completing the proof. $\qquad\square$

