# OpenReview forum: "Controlling Multiple Errors Simultaneously with a PAC-Bayes Bound"
_NeurIPS.cc/2024/Conference — NeurIPS 2024 poster_

### Official Review · Reviewer_fvxp · 2024-07-10

**Soundness:** 4
**Presentation:** 3
**Contribution:** 3
**Rating:** 7
**Confidence:** 3

**Summary:**

The authors suggest an approach for controlling errors of various types (simultaneously) within a PAC-Bayes framework. They derive a high probability bound on the KL divergence between the empirical and distribution risk vectors. This generalizes earlier work in the binary case by Maurer, 2004 and Begin et al 2016. The authors propose a method for using this bound as a (differentiable) training objective via considering a linear combination of the elements of the risk vector and inverting the resulting binary KL divergence.

**Strengths:**

- The main result seems to be a creative and potentially useful extension of the PAC-Bayesian framework.
- Results are generally introduced with some explanation, and given more context after the result, which improves readability.

**Weaknesses:**

- The framework considered would be better motivated by providing a specific example (ideally with an experiment) indicating the usefulness of the method (and illustrating why existing approaches with a union bound are insufficient).
- The authors included the NeurIPS checklist from last year (and not from this year).
- The proof of theorem 2 is very long and either an outline/proof sketch should be provided at the beginning, or the proof should be broken into lemmas/propositions to improve readability.

**Questions:**

- Is it really not possible to upper bound arbitrary linear combinations of different types of risk with a union bound? I would think that bounding finitely many would place linear constraints on other linear combinations, allowing at least some bounds to be derived. It isn’t immediately obvious to me how sharp the resulting bounds would be, or if indeed this approach leads to non-vacuous bounds, but I’d appreciate the authors commenting on this and adding some discussion on the topic if the answer to my question is that it is possible.
- How would $\ell$ be chosen in practice when defining the objective function? It seems the whole purpose of the framework is to provide useful bounds simultaneously for many types of predictive errors. But it also seems the choice of $\ell$ will have a large impact on how well a method trained with proposed approach will perform on tasks that weight certain types of errors heavily.

**Limitations:**

- The authors acknowledge limited experiments as a limitation of the current work. I agree that this is a limitation. In particular, I wonder whether similar results can be obtained using held-out data, and whether or not the bounds derived this way are competitive with this approach. In the one experiment provided, half of MNIST was used to define the prior, while have was used for optimization of the bound. It seems possible that a similar breakdown of data between a training and validation set (trying directly via cross-entropy loss) might lead to similar model performance and tighter bounds on the risk (as in the cases considered in Foong et al). It isn’t immediately obvious whether it is easy to directly invert the multinomial CDF to derive tight bounds on various datasets as in the binary case using holdout data. But I’m curious to hear the authors’ thoughts on this approach.
- The authors point out that the bound is limited to discrete cases. This seems reasonable, and there are many discrete problems where the bound might be useful. I think finding a concrete problem (dataset, with a plausible decision depending on balancing error types) where this bound might be useful would be more convincing to address this limitation than extending the work to continuous problems.

Minor points:

- Line 147: $S_{m,n}^{>0}$ isn’t the interior of $S_{m,n}$ in a topological sense, assuming $S_{m,n}$ is considered with the discrete topology, which seems the natural choice. Otherwise, I don’t know what is meant by interior.
- Line 198: "Different different"

---

> ### Author Rebuttal · Authors · 2024-08-07
>
> Thank you so much for your thorough review and many insightful comments! We are especially glad you find our results creative and the paper well-written.
>
> You mention the paper would be better motivated by an explicit example showing the utility of our method above existing ones. We did not initially believe a comparison to existing methods could be made, since
> - existing methods bound simply the error rate (lumping all errors together), or (via a union bound), bound each error probability, whereas;
> - our method bounds the whole distribution of errors, allowing us to derive bounds on every linear combination of the error probabilities.
>
> However, your (and the other reviewers') insightful comments have clarified that with bounds on the individual error probabilities one can in fact derive bounds on linear combinations of them. Thank you for this astute observation, which enables direct comparison of our result to existing ones! Thus we conducted two quick experiments to demonstrate the utility of our method above existing results.
>
> First, as a quick illustration that our method can beat a union bound, we construct a synthetic example. We take the number of error types to be $M = 25$ (corresponding to one error type for each entry of a $5 \times 5$ confusion matrix), $\delta = 0.05$, and suppose that the empirical risk vector is $\mathbf{R}_S(Q) = (1/25, \dots, 1/25) \in \mathbb{R}^{25}$ and $KL(Q|P) = 0$. We then sample $10^8$ points $\mathbf{p}$ uniformly from the simplex. By counting what proportion of the sampled $\mathbf{p}$ could be values of $\mathbf{R}_D(Q)$ compatible with the union bound and our bound respectively, we obtain the following estimates (with 95% confidence intervals) for the volumes of the two confidence regions:
>
> | $M$   | $m$  | Vol. Union CR                      | Vol. Our CR                        |
> |-------|------|--------------------------------|----------------------------------|
> |       | 100  | 0.3191 (0.3190, 0.3192)            | **0.1758** (0.1757, 0.1758)               |
> | $5^2$ | 300  | 1.313e-3 (1.306e-3, 1.320e-3)        | **3.710e-4** (3.672e-4, 3.748e-4)           |
> |       | 1000 | 5.665e-8 (1.090e-8, 1.024e-7)          | **3.7336e-8** (2.422e-9, 7.225e-8)           |
>
> One can see that our confidence region has a smaller volume in all three cases tested (although in the third case the confidence intervals overlap).
>
> Second, we conducted a more realistic experiment using the HAM10000 dataset, a dataset of images of cancerous and non-cancerous skin marks. We obtain an empirical risk vector $\mathbf{R}_S(Q) = (0.7718, 0.0548, 0.1735)$ for error types $E_0 = $ correct, $E_1 = $ false negative, and $E_2 = $ false positive, by employing our Theorem 2 to do simultaneous minimisation of the Type I and Type II errors with a loss vector $\ell = (0, 1, 2)$, weighting Type II errors as twice as bad as Type I errors (more time would allow professional estimates of the loss vector to be obtained). Again, the results are positive, showing our confidence region to be smaller.
>
> | $M$   | $m$  | Vol. Union CR                      | Vol. Our CR                        |
> |-------|------|--------------------------------|----------------------------------|
> |  3     | 500  | 0.03067 (0.03056, 0.03077)            | **0.02985** (0.02975, 0.02996)|
>
> Another excellent observation you make is that the choice of the loss vector may have a large impact on the training procedure and thus the types of errors the final classifier is liable to make. Unfortunately the choice of a scalar metric to optimise cannot be avoided, the question is just which one to choose. We chose the scalar metric to be the "total risk" $\ell \cdot \mathbf{R}_D(Q)$ for a given $\ell$ because that seems the most natural; while one might be uncertain about the future cost of different error types (for example new medical knowledge may change the relative severity of a type II error in skin cancer diagnosis), one presumably has a rough idea. Our bound then in essence gives you the most assurance with respect to this loss vector and ones close to it. Nevertheless, one can always default to putting a uniform weighting across all error types (except correct classification, where you should pick 0). Do let us know if this makes sense!
>
> You raise the point of calculating a bound using held-out data and mention the very interesting work of Foong et al. It is indeed the case that PAC-Bayes bounds in the literature are usually looser than test set bounds. Given more time we will make this comparison; it is an important and frequently neglected one in the PAC-Bayes literature. Nevertheless, PAC-Bayes bounds, including ours, are worth pursuing for two reasons:
> - First, while the optimal test set bound is known (the Binomial tail bound), PAC-Bayes bounds are still an open research direction and the community has seen a dramatic improvement in the tightness of PAC-Bayes bounds. There is a hope that PAC-Bayes bounds will eventually beat test-set bounds, and we hope that our bound is a stepping stone along this path; by showing how to generalise existing results, it may provide a recipe to generalise any future tighter PAC-Bayes bounds.
> - Second, at least half the value of PAC-Bayes bounds lies in shedding light on generalisation rather than getting tight empirical bounds. For example the framework can help answer questions on sample complexity, as in "Pac-bayes, mac-bayes and conditional mutual information: Fast rate bounds that handle general VC classes" by (Grunwald, Steinke, Zakynthinou, 2021). Our result makes progress on the sample complexity of bounds on the the kl(), relating it to the number of error types.
>
> Finally, thank you for your two minor points! We are grateful that you read the paper so carefully and took the time to relay even these minor points to us, which we have now corrected. We hope we have managed to address all of your questions and that you may even consider raising your score.

---

> > ### Comment · Reviewer_fvxp · 2024-08-12
> >
> > The authors have addressed the primary concerns raised in my review. I have increased my score by a point, as I think the paper should be accepted.

---

> > > ### Author Response · Authors · 2024-08-13
> > >
> > > We are glad we have addressed the important points you raised, and to read that you believe the paper should be accepted. Thank you again for taking the time to write such a thorough review, and for your suggestions and insights which we believe have allowed us to improve the paper!

---

### Official Review · Reviewer_4oQc · 2024-07-11

**Soundness:** 3
**Presentation:** 3
**Contribution:** 3
**Rating:** 7
**Confidence:** 3

**Summary:**

The notion of a set of "error-types", introduced by this work, is a user-defined partition of the product space of predictions and responses, generalizing well-known summaries of the erring behavior of predictors. This work presents a PAC-Bayes bound on the divergence between and empirical and true distributions over a set of "error types" for a choice of posterior prediction scheme. The phrasing of the bound in terms of KL divergence naturally generates bounds on arbitrary linear combinations of the true error type probabilities, which hold simultaneously. The authors present a training objective founded on the minimization of the worst-case value of a pre-defined weighting of error types consistent with the empirical error-type profile of a given choice of posterior over predictors.

**Strengths:**

The paper strikes a good balance between practicality and theory. The problem setting is well-motivated, and the generalization of the confusion matrix through error types allows for the use of the bounds in a variety of non-standard settings. The PAC-Bayes bound is novel.  While PAC-Bayes has been previously used to control the spectral norm between between empirical and true confusion matrices, the bounding distributions over error types in terms of KL divergence straightforwardly generates bounds on the expected cost (over the true distribution over error types) which hold simultaneously, something of notable practical use. The paper is generally well-written.

**Weaknesses:**

One of the main strengths of this work the simultaneous control of arbitrary linear combinations of the error type probabilities. However, the ability of the previous literature to generate some semblance of these guarantees is not particularly well-exposited. Having not previously read any of this line of work, I think an explicit example of how one can construct a bound on a single linear combination might do the paper some good (and might motivate this work past the simultaneous control of all linear combinations, given that the task seems somewhat tricky given a spectral norm bound).

**Questions:**

Is there a quick example of to control a single linear combination of confusion matrix probabilities using a previous bound?
In line 331, what is an "empirically tighter" bound?

**Limitations:**

Yes

---

> ### Author Rebuttal · Authors · 2024-08-07
>
> Thank you for your kind review, especially for noting that our approach is much more flexible than that of Morvant et al. (2012), as one is not limited to the confusion matrix and can instead consider arbitrary user-specified error types.
>
> As for your question on previous results, we have now run an experiment to compare our bound to that of Morvant et al. (2012). Since their result considers the full confusion matrix, a fair comparison requires requires choosing $M = L^2$ error types, where $L$ is the number of labels. To compare, we estimated the volume of our confidence region, which is a kl-ball, against the volume of Morvant's confidence region, which is a spectral-norm-ball. Our Monte Carlo estimates show that Morvant's result is either not applicable or their confidence region is much larger than ours and essentially takes up the entire simplex, hence the volume estimate of 1.000 in the below table. The reason their bound is sometimes inapplicable is because it requires every class to contain at least $8L$ instances (in the $L=5, M=25, m=100$ case this would require each class to contain at least $5 \times 8 = 40$ instances which is impossible with $m=100$ samples). Here the table of the results:
>
> | $M$   | $m$  | Vol. Morvant CR                 | Vol. Our CR                        |
> |-------|------|--------------------------------|----------------------------------|
> |       | 100  | NA ($m$ too small)            | **0.1758** (0.1757, 0.1758)               |
> | $5^2$ | 300  | 1.00  (1.000, 1.000)        | **3.710e-4** (3.672e-4, 3.748e-4)           |
> |       | 1000 | 1.00 (1.000, 1.000)          | **3.7336e-8** (2.422e-9, 7.225e-8)           |
>
> This comparison of the volumes of our confidence region and Morvant's clearly shows that bounds on *linear combinations* of error probabilities derived from our bound will be much tighter that Morvant's, which will be almost as large as they theoretically can be and therefore provide no utility. We were surprised to see that Morvant's highly interesting theoretical bound unfortunately performs so poorly empirically. It is certainly a comparison we will add to the final paper if it is accepted, and we thank you for the very excellent advice which we believe improves the exposition of our result.
>
> On your advice, we are in the process of adding an explicit example of how to calculate a bound on a linear combination using our Theorem 2 and using Morvant's bound. We agree with your observation that this would clarify the import of our quite technical theorem.
>
> Once again we kindly thank you for taking the time to carefully read our paper and give highly constructive and helpful feedback! We believe this has improved the exposition of our paper and helped elucidate our contributions.

---

> > ### Comment · Reviewer_4oQc · 2024-08-10
> >
> > Thanks for further elucidation via experiment. I retain my previous score.

---

> ### Author Response · Authors · 2024-08-13
>
> We are pleased to read that you found the experiment clarifying! Thank you again for taking the time to write a careful review and for your insightful comments, which we believe have improved the work.

---

### Official Review · Reviewer_cfTA · 2024-07-12

**Soundness:** 4
**Presentation:** 3
**Contribution:** 2
**Rating:** 6
**Confidence:** 3

**Summary:**

This paper introduces a novel PAC-Bayes bounds that extend classical kl-based bounds to vector-valued losses that can control several error-types simultaneously. The bound is converted into a differentiable minimization objective and details for the practical implementation of the bound are provided in the Appendix.

**Strengths:**

1- The paper is clearly written and theoretically solid.

2- The contributions have a potential impact in scenarios where simultaneous control over multiple error types is needed.

**Weaknesses:**

1- The proof techniques and the main contribution (Proposition 4) are straightforward extensions of scalar kl-bounds to the multidimensional case. I fear this contribution alone might not be significant enough.

2- The experiment in Section 7 is too simple, further empirical evaluation of the bounds (e.g. simultaneous minimization of Type I and Type II errors in more realistic scenarios) would be very beneficial for the impact of the paper.

3- Wu, Y. S., & Seldin, Y. (2022). Split-kl and PAC-Bayes-split-kl inequalities for ternary random variables. Advances in Neural Information Processing Systems, 35, 11369-11381. is significant related work not discussed in the paper.


In general, I don't think this is a bad paper, but the theoretical contributions are not very original and I feel that the paper needs more experimental backup to be a well-rounded contribution.

**Questions:**

See weaknesses.

**Limitations:**

The limitations are properly discussed in Section 6.

---

> ### Author Rebuttal · Authors · 2024-08-06
>
> Thank you kindly for taking the time to carefully review our paper. Thank you especially for pointing out the unfortunate omission of Wu, Y. S., & Seldin, Y. (2022) from our related work section! The paper is indeed very related and we have now incorporated a discussion of it into our paper.
>
> As for the main contributions of our paper, we emphasise that these are **Theorems 1 and 2**, not Proposition 4. We hope that you take this into consideration with your final score as we feel that Proposition 4 represents a small portion (20%?) of our theoretical contribution. Indeed, as you point out, the proof technique for Proposition 4 is similar to existing results and comes to one page, whereas the techniques required for Theorems 1 and 2 are extensive, novel, and require an additional 12 pages of proof, mostly found in the appendices.
>
> The meat of our paper is:
>   - **Theorem 1** This specialises Proposition 4 to the $d(\cdot, \cdot) = kl(\cdot∥\cdot)$ case and then slightly loosens it to a tractable form so that it can be easily evaluated. The proof is in two parts:
>     - Proving Corollary 1 (lines 286 - 299) by specialising Proposition 4 to the $d(\cdot, \cdot) = kl(\cdot∥\cdot)$ and applying some combinatoric arguments;
>     - Proving Theorem 1 (lines 304 - 316) by loosening the bound to a tractable form. Because we want to keep the bound as tight as possible, we expend some effort in loosening it as little as possible. For this require the technical Lemma 3, which itself requires two helping Lemmas found in Appendix 8.3. These three Lemmas require a further two pages to prove. We split this up into chunks for readability, but appreciate that this might obfuscate what the main contributions are. Hopefully this clears it up with respect to Theorem.
>   - **Theorem 2** This constructs a differentiable training objective from the bound given in Theorem 1. Its proof requires five pages and is found in Appendix 8.5 that does not follow the lines of any existing proof we are aware of.
> We hope you can agree that altogether this represents a non-trivial contribution!
>
> Thank you for highlighting, as we should have realised, that the binarised MNIST dataset we used is not particularly realistic to our stated use-case. On your advice, we conducted an experiment using the HAM10000 dataset, a dataset of images of cancerous and non-cancerous skin marks. We obtain an empirical risk vector $\mathbf{R}_S(Q) = (0.7718, 0.0548, 0.1735)$ for error types $E_0 = $ correct, $E_1 = $ false negative, and $E_2 = $ false positive, by employing our Theorem 2 to do simultaneous minimisation of the Type I and Type II errors with a loss vector $\ell = (0, 1, 2)$, weighting Type II errors as twice as bad as Type I errors (more time would allow professional estimates of the loss vector to be obtained). We will be able to write up the experiment in full in the camera-ready version if accepted, but the results are positive; a Monte Carlo estimate shows that the resulting Confidence Region for $\mathbf{R}_D(Q)$ takes up only $3\%$ of the simplex, which compares favourably to the Confidence Region formed by taking a union bound. Here is a table of results, showing the volume of the confidence regions normalised by the volume of the simplex (in brackets are the confidence intervals from the MC estimates):
>
> | $M$   | $m$  | Vol. Union CR                      | Vol. Our CR                        |
> |-------|------|--------------------------------|----------------------------------|
> |  3     | 500  | 0.03067 (0.03056, 0.03077)            | **0.02985** (0.02975, 0.02996)|
>
> Do please let us know whether you find the new setting of the HAM10000 dataset to be more realistic, and any further details you would like to know!
>
> Thank you again for investing the time to review our paper and provide very helpful feedback which we believe has improved the presentation of the paper and the convincingness of our results.

---

> > ### Comment · Reviewer_cfTA · 2024-08-11
> >
> > Thank you for your clarifying comments, with my doubts regarding the theoretical contributions solved, having read the rest of the rebuttals and with more realistic experiments, I am happy to increase my rating to 6. Including these extra experiments in the camera-ready version will make for a solid contribution.

---

> > > ### Author Response · Authors · 2024-08-13
> > >
> > > We are glad that you found the additional experiment useful! Thank you for raising your score in light of this, and for your suggestions which we believe have improved the work.

---

### Official Review · Reviewer_R3U1 · 2024-07-12

**Soundness:** 3
**Presentation:** 3
**Contribution:** 3
**Rating:** 5
**Confidence:** 4

**Summary:**

In the standard statistical learning theory control of generalization
error typically means studying deviations of the empirical risk from
the risk (for instance, with high probability over the sample).  In
the language of statistics such control doesn't make a difference
between type I & II errors.  This paper goes beyond as it looks at a
multi-objective risk, i.e. 'deviation' of the vector or M empirical
risks from the corresponding risks.  In the context of the paper, the
vector of empirical risks lives on a probability simples and the paper
shows how to control a KL divergence between such a vector and a risk
probability vector.  Then, the paper goes on and studies a PAC-Bayes
formulation of this problem (Theorem 1).  In section 4 the paper
proposes how to construct a differentiable objective for this problem.

**Strengths:**

This is an important problem, especially in the context of
multiobjective errors which are commonplace in practice.  To some
extent these results also extend results of Maurer (2004) (PAC-Bayes
bounds on little-kl) to simpleces (while original results were shown
for Bernoulli case).

**Weaknesses:**

The price for having such a multiobjective setting is M log(M),
whereas the standard union bound would lead to a log(M) cost --- note
that there is a (significant) gap.  However, it should be noted that
this would give a bound on a different objective (not kl() between
points on the simplex).  I think that the paper should argue why such
kl() is interesting or at least design analytical instances and/or
some experiments which would demonstrate that there exist instances
where kl() leads to a better/tigther results (perhaps a performance on
a downstream task?).

**Questions:**

Not a question really, but I'd be interested to hear authors' thoughts
on the weakness against a union bound.  E.g. how would one approach
this comparison problem / experiment design.

**Limitations:**

Seems to be adequate.

---

> ### Author Rebuttal · Authors · 2024-08-06
>
> Thank you for your thoughtful review and for taking the time to give constructive feedback on our work! We are especially grateful for your insight that it is difficult to compare our bound against the standard union bound since we have bounded a different quantity - the vector rather than scalar kl. The reason we did not initially make a comparison was because it seemed to us that the standard union bound was not suitable for our use-case; we are considering a situation such as outlined in paragraph 3 of the introduction, where one cannot anticipate the future costs of certain error types, such as the type I & type II errors you mention. In such a case, a union bound would simultaneously bound the type I & II errors individually, whereas our bound simultaneously bounds *every linear combination* of these two errors. This gives greater assurance when the costs of different error types may change (for example with better medical knowledge), and may be especially useful when there are more than two ways of mis-classifying data.
>
> Your and the other reviewers' insightful comments however, have helped elucidate the fact that a union bound applying to each error probability individually can indeed be used to bound linear combinations of errors, and thereby also constrains the entire vector of error probabilities. For this reason both our bound and a union bound form confidence regions around the empirical risk vector $\mathbf{R}_S(Q)$, and these regions can be compared. One way to do this is to compare their volume, where a smaller volume indicates greater control over the true risk vector $\mathbf{R}_D(Q)$.
>
> As a quick experiment, we take the number of error types to be $M = 25$ (corresponding to one error type for each entry of a $5 \times 5$ confusion matrix), $\delta = 0.05$, and suppose that the empirical risk vector is $\mathbf{R}_S(Q) = (1/25, \dots, 1/25) \in \mathbb{R}^{25}$ and $KL(Q|P) = 0$. We then sample $10^8$ points $\mathbf{p}$ uniformly from the simplex. By counting what proportion of the sampled $\mathbf{p}$ could be values of $\mathbf{R}_D(Q)$ compatible with the union bound and our bound respectively, we obtain the following estimates (with 95% confidence intervals) for the volumes of the two confidence regions (normalised by the volume of the simplex):
>
> | $M$   | $m$  | Vol. Union CR                      | Vol. Our CR                        |
> |-------|------|--------------------------------|----------------------------------|
> |       | 100  | 0.3191 (0.3190, 0.3192)            | **0.1758** (0.1757, 0.1758)               |
> | $5^2$ | 300  | 1.313e-3 (1.306e-3, 1.320e-3)        | **3.710e-4** (3.672e-4, 3.748e-4)           |
> |       | 1000 | 5.665e-8 (1.090e-8, 1.024e-7)          | **3.7336e-8** (2.422e-9, 7.225e-8)           |
>
> One can see that our confidence region has a smaller volume in all three cases tested (although in the third case the confidence intervals overlap). We can therefore see that **the cost to the bound of Mlog(M), rather than log(M) for the union bound, can indeed be outweighed by the fact that we are bounding a different quantity.** We thus outperform the union bound method.
>
> Thank you again for taking the time to carefully review our work, and giving us feedback we believe has enabled us to strengthen the presentation of our result!
>
> We hope that our demonstration that our bound on the kl() can lead to tighter bounds than the union bound approach convinces you of the utility of our result, and that if so you will kindly consider raising your score.

---

> > ### Comment · Reviewer_R3U1 · 2024-08-13
> >
> > Thanks for your explanations, and empirical evaluations which are indeed interesting. I guess, for the final revision it would be interesting to see a more complete empirical suite (even if it's synthetic) to see where the proposed approach "breaks" and the union bound becomes better. Otherwise it is hard to see a full picture.
> >
> > Overall, I think this is interesting and proofs have some new ideas, so I'm in favor of accepting this paper.

---

> ### Author Response · Authors · 2024-08-13
>
> We are glad you found the evaluations interesting and are pleased that you are in favour of accepting the paper! We will indeed expand the evaluations in the final version if it is accepted. Thank you for your detailed review and suggestions, which we believe have improved the paper!

---

### Author Rebuttal · Authors · 2024-08-07

We genuinely thank all four reviewers for their many insightful comments, constructive feedback, and astute observations. We have responded to all four reviewers individually. We hope they let us know in the discussion phase if there is anything we should clarify, or any further results they would like to see, and we will do our best to get back to them promptly. Thank you again investing the time to write these thorough reviews.

---

### Decision · Program_Chairs · 2024-09-25

**Decision:**

Accept (poster)

**Comment:**

The paper cleverly extends the PAC-Bayesian theory. Even if it builds on previous works, the theoretical contributions are not straightforward and require several non-trivial steps.

Of note, the reviews have led the authors to commit to further experiments, which will enrich the manuscript.

The authors should cite Wu and Seldin (2022) as mentioned by Reviewer cfTA.  I also enjoin the authors to mention that the multinomial distribution has been used by Lacasse et al. (2006, Theorem 6) to simultaneously bound the so-called *expected disagreement*, *expected joint success* and *expected joint error* of the C-bound. Even if the latter pursued a different goal and is restricted to a multinomial distribution of solely three events, some proof techniques are similar to the current work. This might be more obvious in the detailed treatment of Germain et al. (2015, Section 5.4).

### References
Lacasse, Laviolette, Marchand, Germain, Usunier. PAC-Bayes Bounds for the Risk of the Majority Vote and the Variance of the Gibbs Classifier. NIPS 2006

Germain, Lacasse, Laviolette, Marchand, Roy. Risk bounds for the majority vote: from a PAC-Bayesian analysis to a learning algorithm. JMLR (2015)